

# Aircraft measurements of High Arctic springtime aerosol show evidence for vertically varying sources, transport and composition

Megan D. Willis[1], Heiko Bozem[2], Daniel Kunkel[2], Alex K.Y. Lee[3], Hannes Schulz[4], Julia Burkart[5], Amir A. Aliabadi[6], Andreas B. Herber[4], W. Richard Leaitch[7], and Jonathan P.D. Abbatt[1]

[1]University of Toronto, Department of Chemistry, Toronto, Ontario, Canada
[2]Johannes Gutenberg University of Mainz, Institute for Atmospheric Physics, Mainz, Germany
[3]National University of Singapore, Department of Civil and Environmental Engineering, Singapore
[4]Alfred Wegener Institute Helmholtz-Center for Polar and Marine Research Bremerhaven, Bremerhaven, Germany
[5]University of Vienna, Faculty of Physics, Aerosol Physics and Environmental Physics, Vienna, Austria
[6]School of Engineering, University of Guelph, Guelph, Ontario, Canada
[7]Environment and Climate Change Canada, Toronto, Ontario, Canada

**Correspondence:** Megan D. Willis (megan.willis@mail.utoronto.ca)

**Abstract.** The sources, chemical transformations and removal mechanisms of aerosol transported to the Arctic are key factors that control Arctic aerosol-climate interactions. Our understanding of sources and processes is limited by a lack of vertically resolved observations in remote Arctic regions. We present vertically resolved observations of trace gases and aerosol composition in High Arctic springtime, made largely north of 80°N, during the NETCARE campaign. Trace gas gradients observed on

these flights defined the polar dome as north of $66 - 68.5°$N and below potential temperatures of $283.5 - 287.5$ K (Bozem et al., 2018). In the polar dome, we observe evidence for vertically varying source regions and chemical processing. These vertical changes in sources and chemistry lead to systematic variation in aerosol composition as a function of potential temperature. We show evidence for sources of aerosol with higher organic aerosol (OA), ammonium (NH4) and refractory black carbon (rBC) content in the upper polar dome. Based on FLEXPART-ECMWF calculations, air masses sampled at all levels inside

the polar dome (i.e., potential temperature $< 280.5$ K, altitude $< \sim 3.5$ km) subsided during transport over transport times of at least 10 days. Air masses at the lowest potential temperatures, in the lower polar dome, had spent long times ($>10$ days) in the Arctic, while air masses in the upper polar dome had entered the Arctic more recently. These differences in transport history were closely related to aerosol composition. In the lower polar dome, the measured sub-micron aerosol mass was dominated by sulphate (mean 74%), with lesser contributions from rBC (1%), NH4 (4%) and OA (20%). At higher altitudes and warmer

potential temperatures, OA, NH4 and rBC contributed 42%, 8% and 2% of aerosol mass, respectively. A qualitative indication for the presence of sea salt showed that sodium chloride contributed to sub-micron aerosol in the lower polar dome, but was not detectable in the upper polar dome. Our observations suggest that long-term, surface-based measurements underestimate the contribution of OA, rBC and NH4 to aerosol transported to the Arctic troposphere in spring.



# 1 Introduction

Arctic regions are warming faster than the global average, with significant impacts on local ecosystems and local people
(e.g, Bindoff et al., 2013; Hinzman et al., 2013). While Arctic warming is driven largely by increasing concentrations of
anthropogenic greenhouse gases and local feedback mechanisms, short-lived climate forcing agents also impact Arctic climate.
In particular, short-lived species such as aerosol, tropospheric ozone and methane, are important climate forcers (e.g., Law
and Stohl, 2007; Quinn et al., 2008). The impact of pollution aerosol, transported northward over long distances, on Arctic
climate has been significant. For example, a large fraction of greenhouse gas induced warming ($\sim$60%) has been offset by
anthropogenic aerosol over the past century, such that reductions in sulphur emissions in Europe since 1980 can explain a large
amount of Arctic warming since that time ($\sim$0.5 K) (Fyfe et al., 2013; Najafi et al., 2015; Navarro et al., 2016). These estimates
are compelling, and at the same time global models that form the basis of our predictive capability often struggle to reproduce
key characteristics of Arctic aerosol, such as the seasonal cycle and vertical distribution (Shindell et al., 2008; Emmons et al.,
2015; Monks et al., 2015; Eckhardt et al., 2015; Arnold et al., 2016). Our incomplete understanding of Arctic aerosol processes
results in diverse and frequently poor model skill in simulating Arctic aerosol both at the surface and through the troposphere,
and therefore also in accurately simulating aerosol-climate interactions (Arnold et al., 2016). This challenge arises in part due
to a lack of vertically-resolved observations in Arctic regions.

Particle composition drives aerosol optical properties (e.g., Boucher and Anderson, 1995; Jacobson, 2001; Wang et al.,
2008), ice nucleation efficiency (e.g., Abbatt et al., 2006; Hoose and Möhler, 2012), and heterogeneous chemistry that impacts
both gas and particle composition (e.g., Fan and Jacob, 1992; Mao et al., 2010; Abbatt et al., 2012). The vertical distribution
of aerosol and its chemical and physical properties can impact Arctic regional climate in a number of ways. First, absorption
of incoming solar radiation by aerosol (e.g., black carbon) can lead to warming in the lower troposphere when present near
the surface. In contrast, absorbing aerosol present at higher altitudes causes cooling at the surface and impacts atmospheric
stratification (Rinke et al., 2004; Ritter et al., 2005; Treffeisen et al., 2005; Shindell and Faluvegi, 2009; Engvall et al., 2009).
Further, location in the troposphere impacts deposition to high albedo surfaces, depending on the mechanism of removal
(e.g., Macdonald et al., 2017). Absorbing aerosol deposited to the surface has a strong impact on the albedo of ice and snow,
efficiently leading to warming (Clarke and Noone, 1985; Hansen and Nazarenko, 2004; Flanner et al., 2009; Flanner, 2013).
Second, neutralization of acidic sulphate impacts aerosol water content and aerosol phase, with implications for the magnitude
of aerosol-radiation interactions (Boucher and Anderson, 1995; Wang et al., 2008). Third, sulphuric acid coatings on particles
can decrease their ability to act as ice nucleating particles (INPs), leading to larger, more readily precipitating ice crystals. This
process can lead to enhanced atmospheric dehydration, leading to diminished long-wave forcing (Curry and Herman, 1985;
Blanchet and Girard, 1994, 1995). Further, particles containing mineral dust, organic species, sea salt or neutralized sulphate
can increase the number of ice crystals at cirrus temperatures, also leading to impacts on long-wave and short-wave cloud
forcing (Sassen et al., 2003; Abbatt et al., 2006; Baustian et al., 2010; Wagner et al., 2018). Finally, Arctic pollution aerosol
can impact the micro-physical properties of liquid-containing clouds, by increasing liquid water path and decreasing droplet



radius. Such micro-physical changes can result in enhanced long-wave warming effects during winter and spring (Garrett and Zhao, 2006; Lubin and Vogelmann, 2006; Zhao and Garrett, 2015).

Observations at Arctic ground-based monitoring stations form the basis of our current knowledge about Arctic aerosol seasonality, chemical composition and sources. These long-term observations have demonstrated a pronounced seasonal cycle

in Arctic aerosol mass concentrations, particle size distribution and composition, driven by seasonal variations in northward long range transport and aerosol wet removal (e.g., Quinn et al., 2007; Sharma et al., 2013; Tunved et al., 2013; Croft et al., 2016; Asmi et al., 2016; Nguyen et al., 2016; Freud et al., 2017; Leaitch et al., 2018). Aerosol mass concentrations peak in winter to early spring when long range transported accumulation mode particles (200 – 400 nm mode diameter), referred to broadly as "Arctic Haze", dominate the particle size distribution (e.g., Croft et al., 2016; Freud et al., 2017). A mixture

of natural and anthropogenic aerosol is transported to Arctic regions by near-isentropic transport along surfaces of constant potential temperature that slope upwards toward the Arctic (Klonecki et al., 2003; Stohl, 2006). The sloping isentropic surfaces form a closed "dome" over the polar region; this polar dome is zonally asymmetric, extends further south in winter and contracts northward in spring to summer (Shaw, 1995; Law and Stohl, 2007; Stohl, 2006; Law et al., 2014). Arctic Haze observed near the surface is largely acidic sulphate, with lesser contributions from organic aerosol, dust, nitrate, ammonium and sea salt (Li

and Winchester, 1989; Quinn et al., 2002, 2007; Shaw et al., 2010; Leaitch et al., 2018). Aerosol acidity increases during winter and reaches a peak in late spring (Sirois and Barrie, 1999; Toom-Sauntry and Barrie, 2002), before the return of wet removal brings the Arctic toward near-pristine conditions with more neutralized aerosol (Engvall et al., 2008; Browse et al., 2012; Breider et al., 2014; Wentworth et al., 2016; Croft et al., 2016). Sea salt is thought to be an important contributor to Arctic Haze in winter to early spring owing to stronger wind speeds over nearby oceans, potential wind-driven sources in ice

and snow-covered regions, and open leads (Leck et al., 2002; Shaw et al., 2010; May et al., 2016; Huang and Jaegle, 2017; Kirpes et al., 2018). The major source region of near-surface Arctic Haze in winter and early spring is Northern Europe and Northern Asia/Siberia, but the magnitude of sources in this region have been decreasing in recent decades (Barrie and Hoff, 1984; Sharma et al., 2004; Koch and Hansen, 2005; Sharma et al., 2006; Hirdman et al., 2010; Huang et al., 2010b; Gong et al., 2010; Bourgeois and Bey, 2011; Stohl et al., 2013; Sharma et al., 2013; Monks et al., 2015; Qi et al., 2017). Surface-based

observations have provided substantial insight into Arctic aerosol processes, but owing to the stability of the troposphere the surface can be decoupled from the atmosphere above. Therefore, surface-based observations may not represent the overall composition of aerosol transported to the Arctic troposphere (e.g., Stohl, 2006; McNaughton et al., 2011). How transported aerosol present throughout the troposphere is related to Arctic Haze observed near the surface remains an unresolved question (Law et al., 2014; Arnold et al., 2016).

Vertically resolved observations of the Arctic atmosphere, in the last 20 years, have furthered our understanding of the properties, processes, and impacts of Arctic aerosol. Some of the only seasonal airborne observations of aerosol sulphate, suggested that the aerosol seasonal cycle may differ aloft compared to near the surface (Klonecki et al., 2003; Scheuer et al., 2003). Clean-out may begin to take place near the surface in late April to May, before significant changes occur aloft. Intensive observations were made during the International Polar Year (IPY) in 2007–2008. During IPY, high concentrations of aerosol

and trace gases from biomass and fossil fuel burning were observed in discrete layers that did not appear related to Arctic



Haze observed near the surface (e.g., Warneke et al., 2009; Schmale et al., 2011; Brock et al., 2011; Law et al., 2014). Also during IPY, aerosol ammonium content increased from near the surface toward the middle to upper troposphere (Fisher et al., 2011). The largest fraction of sulphate was observed in the lower $\sim 2\,\mathrm{km}$, in general agreement with long term monitoring observations. In years with high burned area in the Northern Hemisphere, such as 2008, biomass burning sources contribute a

significant fraction of black carbon and organic aerosol in the Arctic troposphere (Warneke et al., 2009; Hecobian et al., 2011; McNaughton et al., 2011; Bian et al., 2013). In years with moderate burned area consistent with decadal mean conditions, anthropogenic sources can still lead to enhanced absorbing aerosol in the Arctic mid-troposphere (Liu et al., 2015). IPY observations in the Alaskan Arctic demonstrated that background pollution aerosol (i.e., in air masses with CO $<170\,\mathrm{ppb_v}$) and aerosol in the near surface layer (i.e., in air masses with depleted $O_3$) contained a larger fraction of sulphate compared to

aerosol attributed to biomass or fossil fuel burning (Brock et al., 2011). The properties of Arctic background air masses were generally consistent with median observations at a nearby ground station, Utqiaġvik (Barrow), Alaska (Brock et al., 2011). This background aerosol often has diffuse source regions that are difficult to diagnose precisely using 10 day backward trajectories (e.g, Brock et al., 2011; Qi et al., 2017; Leaitch et al., 2018).

Our knowledge of the vertical distribution of Arctic aerosol source regions has also been extended by recent airborne observa-

tions. Results from modelling efforts generally agree that Arctic pollution aerosol is a result of a combination of anthropogenic and natural sources from mid-latitudes in the Northern Hemisphere; particularly a combination of European, North and South Asian and North American source regions (e.g., Law et al., 2014; Arnold et al., 2016). However, modelling efforts provide less quantitative agreement on the magnitude of the contributions of each region near the surface and as a function of altitude. Our emerging understanding is of Northern Eurasian sources dominating near the surface in winter, while North America and

South/East Asia can be important in the middle to upper troposphere (e.g., Koch and Hansen, 2005; Shindell et al., 2008; Huang et al., 2010a; Law et al., 2014; Liu et al., 2015; Arnold et al., 2016; Qi et al., 2017). In spring, as the polar dome recedes northward, North American and Asian sources become more important at all altitudes (Koch and Hansen, 2005; Fisher et al., 2011; Xu et al., 2017). Overall, more southerly source regions become more important at higher altitudes (e.g., Stohl, 2006; Fisher et al., 2011; Harrigan et al., 2011), and the importance of Asian sources above the Arctic surface is being increasingly

recognized (e.g., Koch and Hansen, 2005; Fisher et al., 2011; Xu et al., 2017). The magnitude of Asian influence on the lower troposphere inferred from models in spring varies significantly and depends on emissions estimates and assumptions about aerosol removal efficiency during transport (e.g., Matsui et al., 2011a, b). Source apportionment of recent vertically-resolved Arctic black carbon observations demonstrated that Eastern and Southern Asia make important contributions throughout the troposphere in spring, with a more significant contribution at higher altitudes (Xu et al., 2017). Northern Asia was a more

important source region near the surface (Xu et al., 2017). Changes in source strengths at mid-latitudes and within the Arctic strongly impacts the dominant source regions for different aerosol species (Arnold et al., 2016).

Previous vertically resolved observations of Arctic pollution aerosol frequently focused on episodic events of high pollutant concentrations, largely owing to their potential radiative impact (e.g., Rahn et al., 1977; Engvall et al., 2009; Warneke et al., 2009; Law et al., 2014). We know less about the vertical distribution of Arctic aerosol properties within the High Arctic

polar dome and under conditions consistent with Arctic background conditions (e.g., CO $<170\,\mathrm{ppb_v}$ (Brock et al., 2011)).



Improved understanding of different anthropogenic and natural contributions to Arctic aerosol will provide a scientific basis for sustainable climate mitigation and adaptation strategies. Within the framework of the NETCARE project, airborne observations of Arctic Haze aerosol were made across the North American and European Arctic in April 2015. Observations of trace gas gradients during this campaign were used by Bozem et al. (2018) to define the boundaries of the polar dome. The location of

the maximum trace gas gradient defined the polar dome as north of 66 – 68.5°N and below potential temperatures of 283.5 – 287.5 K. Based on Bozem et al. (2018) we use a conservative definition of the polar dome area based on the interquartile range of the location of maximum trace gas gradient: north of 69.5°N and below 280.5 K. In this work, we quantify vertical changes in sub-micron aerosol composition in the Canadian High Arctic within the boundaries of the polar dome and in the absence of episodic transport events of high pollutant concentrations. Using the Lagrangian particle dispersion model FLEXPART,

we explore the source regions that drive observed sub-micron aerosol in the springtime polar dome. Finally, we examine the depth over which aerosol consistent with surface monitoring observations extends vertically in the polar dome, and assess the representativeness of ground-based observations for aerosol transported to the polar dome in spring.

## 2 Methods

### 2.1 High Arctic measurements

#### 2.1.1 Measurement platform and inlets

Measurements of aerosol, trace gases and meteorological parameters were made in High Arctic spring aboard the Alfred Wegener Institute (AWI) Polar 6 aircraft, an unpressurised DC-3 aircraft converted to a Basler BT-67 (Herber et al., 2008), as part of the Network on Climate and Aerosols: Addressing Key Uncertainties in Remote Canadian Environments project (NETCARE, http://www.netcare-project.ca), and in partnership with the Polar Airborne Measurements and Arctic Regional

Climate Model Simulation Project (PAMARCMiP (Herber et al., 2012)). Measurements on a total of 10 flights took place from 4 - 22 April, 2015 based at four stations along the PAMARCMiP track: Longyearbyen, Svalbard (78.2 ° N, 15.6 ° E); Alert, Nunavut, Canada (82.5 ° N, 62.3 ° W); Eureka, Nunavut, Canada (80.0 ° N, 85.9 ° W); and Inuvik, Northwest Territories, Canada (68.4 ° N, 133.7 ° W). To focus our analysis on aerosol within the polar dome, a subset of six flights in the High Arctic during 7 – 13 April 2015 are considered in this analysis (Figure 1). The vertical extent of these flights is shown in Figure S1.

During measurement flights aircraft speed was maintained at $\sim 75\,\mathrm{m\,s^{-1}}$ ($\sim 270\,\mathrm{km\,h^{-1}}$), with ascent and descent rates of $\sim 150\,\mathrm{m\,min^{-1}}$.

Aerosol and trace gas inlets were identical to those used aboard Polar 6 during the NETCARE 2014 summer campaign, and are described in Leaitch et al. (2016) and Willis et al. (2016). Briefly, aerosol was sampled near isokinetically through a stainless steel shrouded diffuser inlet, with near unity transmission of particles 20 nm to $\sim 1\,\mu\mathrm{m}$ in diameter at typical survey

airspeeds and a total flow rate of approximately $55\,\mathrm{L\,min^{-1}}$. Bypass lines off the main inlet, at angles of 45°, carried aerosol to various instruments. Performance of the aerosol inlet used here was characterized by Leaitch et al. (2016). Aerosol was not





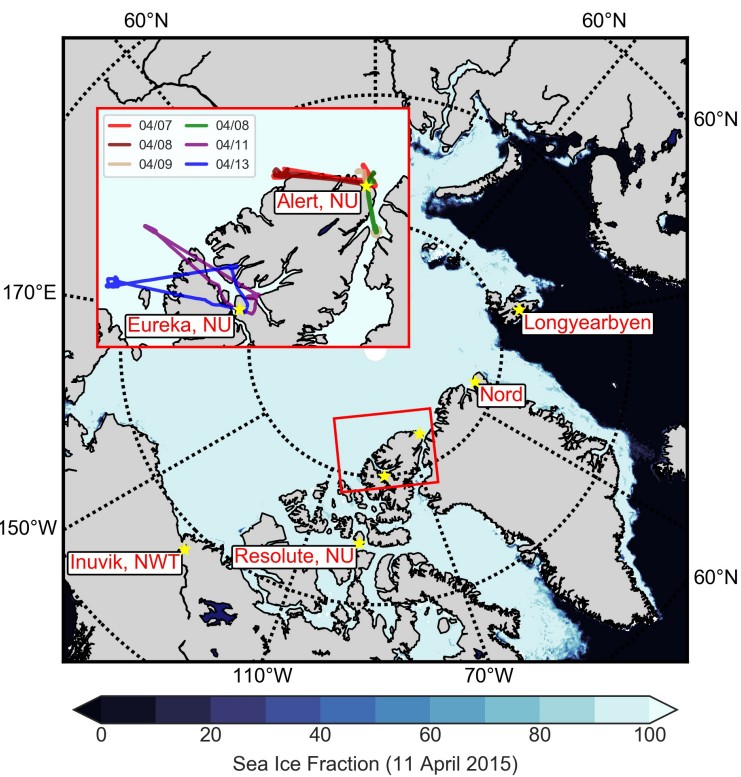

**Figure 1.** Map of the NETCARE 2015 campaign study area, showing sea ice concentrations on 11 April 2015 (Spreen et al., 2008). All stations along the NETCARE/PAMARCMiP 2015 track are shown with red stars (Longyearbyen, Svalbard; Alert, Nunavut; Eureka, Nunavut; Resolute Bay, Nunavut; and Inuvik, Northwest Territories). (Inset) Flight tracks from six flights during 7 – 13 April, 2015 based in Alert and Eureka, Nunavut, which are the focus of this work.

actively dried prior to sampling; however, the temperature in the inlet line within the aircraft cabin was at least 15 ° C warmer than the ambient temperature so that the relative humidity (RH) decreased significantly.

### 2.1.2 State parameters

State parameters and meteorological conditions were measured with an AIMMS-20, manufactured by Aventech Research
5    Inc. (Barrie, ON, Canada; aventech.com/products/aimms20.html). The AIMMS-20 consists of three modules: (1) an Air Data Probe, which measures temperature and the three-dimensional aircraft-relative flow vector (TAS, angle of attack, and side slip) with a three-dimensional accelerometer for measurement of turbulence; (2) an Inertial Measurement Unit, which provides the aircraft angular rate and acceleration; and (3) a Global Positioning System for aircraft three-dimensional position and inertial



velocity. Vertical and horizontal wind speeds are measured with accuracies of $0.75$ and $0.50\,\mathrm{m\,s^{-1}}$ respectively. Accuracy and precision of the temperature measurement are $0.30$ and $0.10\,^{\circ}\mathrm{C}$ respectively. Potential temperature was calculated using temperature and pressure measured by the AIMMS-20.

### 2.1.3 Trace gases

*Carbon Monoxide.* CO concentrations were measured at $1\,\mathrm{Hz}$ with an Aerolaser ultra-fast carbon monoxide monitor (model AL 5002), based on VUV-fluorimetry using excitation of CO at $150\,\mathrm{nm}$. The instrument was modified such that in situ calibrations could be conducted in flight. Measured concentrations were significantly higher than the instrument detection limit. The measurement precision is $\pm1.5\,\mathrm{ppbv}$, with an instrument stability based on in-flight calibrations of 1.7%.

*Water Vapour and Carbon Dioxide.* $H_2O$ and $CO_2$ measurements were made at $1\,\mathrm{Hz}$ using non-dispersive infrared absorption with a LI-7200 enclosed $CO_2/H_2O$ Analyzer from LI-COR Biosciences. In situ calibrations were performed during flight at regular intervals ($15$–$30\,\mathrm{min}$) using a NIST traceable $CO_2$ standard with zero water vapour concentration. Measured concentrations were significantly higher than the instrument detection limit. The measurement precision for $CO_2$ is $\pm0.05\,\mathrm{ppmv}$, with an instrument stability based on in-flight calibrations of 0.5%. The measurement precision for $H_2O$ is $\pm18.5\,\mathrm{ppmv}$, with an instrument stability based on in-flight calibrations of 2.5%.

*Ozone.* $O_3$ concentrations were measured, with a time resolution of $10\,\mathrm{s}$, using UV absorption at $254\,\mathrm{nm}$ with a Thermo Scientific ozone analyzer (model 49i). The measurement uncertainty is $\pm0.2\,\mathrm{ppbv}$.

### 2.1.4 Particle concentrations

Aerosol number size distributions from $100\,\mathrm{nm}$ to $1\,\mathrm{\mu m}$ were acquired with a Droplet Measurement Technology (DMT) Ultra High Sensitivity Aerosol Spectrometer (UHSAS), with a flow rate of $55\,\mathrm{cm^3\,min^{-1}}$ from a bypass flow off the main aerosol inlet. The UHSAS measured scattered light intensity of single particles that cross an intra-cavity $1054\,\mathrm{nm}$ $\mathrm{Nd^{3+}:YLiF_4}$ laser to determine particle size (Cai et al., 2008). We present these observations as the number of particles between $100 - 500\,\mathrm{nm}$ ($N_{100-500}$) and the number greater than $500\,\mathrm{nm}$ ($N_{>500}$). Recent work has highlighted the impact of rapid pressure changes, during aircraft ascent and descent, on reported UHSAS particle concentrations (Brock et al., 2011; Kupc et al., 2018). However, comparison between particle measurements during NETCARE 2015 suggests that these effects are not significant likely owning to the relatively slow vertical speed of the Polar 6 (Schulz et al., 2018).

### 2.1.5 Particle composition

*Refractory Black Carbon.* Concentrations of particles containing refractory black carbon (rBC) were measured with a DMT single particle soot photometer (SP2) (Schwarz et al., 2006; Gao et al., 2007). The SP2 uses a continuous intra-cavity Nd:YAG laser ($1064\,\mathrm{nm}$) to classify particles as either incandescent (rBC) or scattering (non-rBC), based on the individual particle's interaction with the laser beam. The peak incandescence signal is linearly related to the rBC mass. The SP2 was calibrated with Fullerene Soot (Alfa Aesar) standard by selecting a narrow size distribution of particles with a differential mobility analyzer





upstream of the SP2 (Laborde et al., 2012). The SP2 efficiently detected particles with rBC mass of 0.6 to 328.8 fg, which corresponds to $85 - 704\,\mathrm{nm}$ mass equivalent diameter (assuming a void free bulk material density of $1.8\,\mathrm{g\,cm^{-3}}$). rBC mass concentrations were not corrected for particles outside the instrument size range, and the measurement uncertainty is $\pm15\%$ (Laborde et al., 2012). Measurements of rBC during NETCARE 2015 are discussed in detail by Schulz et al. (2018).

*Non-refractory Aerosol Composition.* Non-refractory aerosol composition was measured with an Aerodyne time-of-flight aerosol mass spectrometer (ToF-MS) (DeCarlo et al., 2006). Operation of the ToF-AMS aboard Polar 6 and characterization of the pressure - controlled inlet system is described in Willis et al. (2016) and Willis et al. (2017). The ToF-AMS deployed here was equipped with an infrared laser vaporization module similar to that of the DMT SP2 (SP – laser) (Onasch et al., 2012); however, rBC concentrations during the flights discussed here were generally below ToF-AMS detection limits ($\sim0.1\,\mathrm{ug\,m^{-3}}$

for rBC) so SP2 measurements of rBC are used in this work. The instrument was operated up to an altitude of $\sim3.5\,\mathrm{km}$, and the temperature of the ToF-AMS was passively maintained using a modular foil-lined insulating cover. The ToF-AMS was operated in "V-mode" with a mass range of $m/z$ 3–290, alternating between ensemble mass spectrum (MS) mode for 10 s (two cycles of 5 s MS open and 5 s MS closed) with the SP – laser on, MS mode with the SP – laser off for 10 s, and efficient particle time-of-flight (epToF) mode with the SP – laser on for 10 s (Supplement Table S1). Single particle observations were

made on two flights; this ToF-AMS operation mode is described below. Only observations made with the SP – laser off are used to quantify non-refractory aerosol composition. Filtered ambient air was sampled with the ToF-AMS at least three times per flight, for a duration of at least 5 min, to account for contributions from air signals.

Species comprising non-refractory particulate matter are quantified by the ToF-AMS, including sulphate, nitrate, ammonium, and the sum of organic species. The ToF-AMS is also capable of detecting sea salt (Ovadnevaite et al., 2012). The detection

efficiency of sea salt-containing particles is dependent on not only the ambient RH but also the temperature of the tungsten vaporizer (Ovadnevaite et al., 2012). A quantitative estimate of sea salt mass is not possible with these measurements and this species is not included in calculation of aerosol chemical mass fractions, such that mass fractions presented represent non-refractory aerosol species and rBC measured by the SP2. The vaporizer temperature was calibrated with sodium nitrate particles, and was operated at a temperature of $\sim650°$ C. ToF-AMS signals for sea salt, in particular $\mathrm{NaCl^+}$ (m/z 57.96),

can be used to quantify sea salt (Ovadnevaite et al., 2012); however, here we use the $\mathrm{NaCl^+}$ signal only as a qualitative indication for the presence of sea salt owing to uncertainties in sea salt collection efficiency as a function of RH and the lack of RH measurement in the sampling line. Ammonium nitrate calibrations (Jimenez et al., 2003) were carried out twice during the campaign as well as before and after, owing to restricted access to calibration instruments during the campaign. Air-beam corrections were referenced to the appropriate calibration in order to account for differences in instrument sensitivity

between flights. The relative ionization efficiencies for sulphate and ammonium ($\mathrm{RIE_{SO_4}}$ and $\mathrm{RIE_{NH_4}}$) were $0.9\pm0.1$ and $3.4\pm0.3$. The default relative ionization efficiency for organic species (i.e. $\mathrm{RIE_{Org}} = 1.4$) was used, which is appropriate for oxygenated organic aerosol (Jimenez et al., 2003, 2016). Elemental composition was calculated using the method presented in Canagaratna et al. (2015). Data were analysed using the Igor Pro based analysis tool PIKA (v.1.16H) and SQUIRREL (v.1.57l) (Seuper, 2010). Detection limits and propagated uncertainties (i.e., $\pm$(detection limit + total uncertainty)) for sulphate,

nitrate, ammonium, and organics at a 10 second time resolution were $\pm(0.009\,\mathrm{\mu g\,m^{-3}} + 35\%)$, $\pm(0.001\,\mathrm{\mu g\,m^{-3}} + 33\%)$,





$\pm(0.003\,\mu\mathrm{g\,m^{-3}} + 33\%)$, and $\pm(0.08\,\mu\mathrm{g\,m^{-3}} + 37\%)$, respectively. We note that ion ratios commonly reported from ToF-AMS measurements of ammonium and sulphate are not appropriate for estimating aerosol neutralization (Hennigan et al., 2015), so we do not report these here. A composition-dependent collection efficiency (CDCE) was applied to correct ToF-AMS mass loadings for non-unity particle detection due to particle bounce on the tungsten vaporizer (Middlebrook et al.,

2012), which resulted in a median (quartile range) CE correction of 18% (12 – 28%) applied uniformly to non-refractory aerosol species. ToF-AMS total non-refractory aerosol mass agreed with estimated UHSAS aerosol mass within a factor of two.

   *ToF-AMS Single Particle Measurements.* The ToF-AMS was operated in Event Trigger Single Particle (ETSP) mode on two flights (Table S1). ETSP is run in the single-slit particle-time-of-flight (pToF) mode. A particle event is defined as a single mass

spectrum (MS) extraction or set of consecutive MS extractions associated with a single particle being vaporized and producing MS signals. The number of MS extractions obtained during a particle event is determined by the pulser frequency, and thus the mass range, set during acquisition; in this case 30.9 kHz, corresponding to a pulser period of 32.4 $\mu$s ($m/z$ 3–290). Under these conditions, at least a single mass spectrum is collected per particle event. Saving mass spectra associated with a particle event is triggered in real-time based on the signals present in up to three continuous ranges of mass-to-charge ratios, called

Regions of Interest (ROIs). Three ROIs were used in this work such that a signal above a specified ion threshold in any ROI would trigger saving a mass spectrum (Table S2). Ion thresholds were purposely set low to collect a large number of false positives that are subsequently removed based on the relationship between total aerosol ion signal (i.e., excluding air) and particle size (Figure S2), similar to the approach described in Lee et al. (2015). Two background regions in the particle size distribution (10 – 50 nm and 2000 – 4000 nm) were selected to determine the average background ion signal excluding air

peaks, and particle events considered "real" must be between 80 – 1000 nm with ion signals above the mean background plus three times its standard deviation (Figure S2). A simplified fragmentation table, described in Lee et al. (2015), was applied to particle mass spectra identified as "real" and fragmentation corrections were based on higher mass resolution ensemble MS spectra collected concurrently. A total of 1677 "real" particle spectra were collected over two flights (8 and 13 April, 2015). k-means cluster analysis was applied to particle spectra to explore different particle mixing states, following Lee et al. (2015).

A two-cluster solution was selected to describe the 1677 total "real" particle spectra. Owing to the small number of particle spectra and the lack of specificity in organic aerosol peaks from highly oxygenated aerosol, increasing the number of clusters did not yield physically meaningful information. Mean mass spectra and mass spectral histrograms for each particle class are shown in Figure S2. ETSP data was analysed using the Igor Pro based analysis tools Tofware version 2.5.3.b (developed by TOFWERK and Aerodyne Research, Inc.), clustering input preparation panel (CIPP) version ETv2.1b and cluster analysis

panel (CAP) version ETv2.1 (developed by A.K.Y Lee and M.D. Willis).

## 2.2   Air mass history from particle dispersion modelling

The Lagrangian particle dispersion model FLEXible PARTicle (FLEXPART) (Brioude et al., 2013) driven by meteorological analysis data from the European Center for Medium-Range Weather Forecasts (ECMWF) was used to study the history of air masses prior to sampling during NETCARE flights. The ECMWF data had a horizontal grid spacing of 0.25° and 137 vertical





levels. Here, we use FLEXPART-ECMWF run in backward mode to study the origin of air influencing aircraft-based aerosol and trace gas measurements. Individual FLEXPART parcels were initialized along the flight track every three minutes and then traced back in time for 10 days, providing a time-resolved information on source regions of trace species measured along the flight track. FLEXPART-ECMWF output was provided every three hours over the 10 day period, with horizontal grid spacing

of $0.25°$ and 10 vertical levels. Model output provides a residence time of air prior to sampling that can be used to generate a map of the potential emission sensitivity (PES). We show maps of the PES with units of seconds (i.e., representing an air mass residence time); absolute residence times depend on the model output time step and the extent of spatial averaging. The PES represents integration of model output over a period of time prior to sampling (i.e., 10 days), also referred to as the "time before measurement," and over a vertical range. We show maps of both the total column PES (i.e., $0 - 20\,\mathrm{km}$) and partial column PES

(i.e., $0 - 200\,\mathrm{m}$).

By integrating model output at each model release over specific pressure levels and/or latitude ranges we used FLEXPART-ECMWF to calculate the residence time of air in the middle-to-lower polar dome. The horizontal extent of the polar dome was defined based on Bozem et al. (2018) as north of $69.5°\,\mathrm{N}$. The vertical extent of the middle-to-lower polar dome was defined based on trace gas profiles as below $265\,\mathrm{K}$ ($\sim1550\,\mathrm{m}$). Calculation of this quantity is analogous to calculating the PES (i.e., by

integrating in time and space), with constraints on altitude and location. This residence time is reported as a relative residence time over the 10 day FLEXPART-ECMWF backward integration time. Aircraft observations were sub-sampled to the model time resolution by taking a one-minute average of measurements around the model release time, when the aircraft altitude was within $\pm100\,\mathrm{m}$ of the model release altitude.

## 3 Results and Discussion

### 3.1 Transport regimes in the polar dome

We focus on observations made on six flights in the High Arctic during NETCARE 2015 over the period 7 – 13 April 2015. Figure 1 illustrates flight tracks during this period on a map of the sea ice concentration from 11 April 2015. Observations of trace gas gradients during this campaign defined the region inside the polar dome at north of $69.5°\mathrm{N}$ and below $280.5\,\mathrm{K}$ ($\sim3.5\,\mathrm{km}$) (Bozem et al., 2018). Zonal mean potential temperature cross sections from ECMWF for the period 7 – 13 April

2015 agree generally with this definition of the polar dome, and demonstrates that our observations were made in the coldest air masses present in the Arctic region during this time (Figure S4). CO concentrations observed in the polar dome were consistent with "Arctic background" air masses identified in previous airborne observations and with monthly mean CO concentrations at Alert, Nunavut, Canada (Figure S5). This suggests that our observations during April 2015 in the polar dome were not strongly impacted by episodic transport events of high pollutant concentrations (Brock et al., 2011). We restrict our analysis

to those air masses residing in the polar dome, to determine the sources and processes contributing to aerosol composition within this region during spring. When discussing observations and model predictions, we use potential temperature instead of height or pressure for two reasons. First, the location of the polar dome and transport northward are dictated by potential temperature rather than absolute height. Second, trace gases and aerosol observed in the polar dome varied systematically



with potential temperature, but showed less systematic variability with pressure (Figure S6). Altitude profiles of absolute and potential temperature are shown in Figure S7. In this section, we discuss transport patterns inferred from trace gas observations and FLEXPART-ECMWF air mass history, then in Section 3.2 we discuss observed aerosol composition in the context of these transport patterns.

Trends in trace gas concentrations with potential temperature illustrate different transport regimes within the polar dome (Figure 2). Based on the mean vertical profiles of trace gases, we divided observed vertical profiles into three ranges of potential temperature (Figure 2: $245 - 252\,\mathrm{K}$, $252 - 265\,\mathrm{K}$ and $265 - 280\,\mathrm{K}$) to guide interpretation of air mass history, transport characteristics and aerosol composition in the polar dome. We refer to these three ranges of potential temperature as the lower, middle and upper polar dome, respectively (dashed horizontal lines in Figure 2), and discuss the characteristics of each region

in turn. First, in the coldest and driest air masses ($245 - 252\,\mathrm{K}$), we consistently observed temperature inversion conditions with potential temperature increasing by $37\,\mathrm{K\,km^{-1}}$ compared to $11\,\mathrm{K\,km^{-1}}$ above the lower polar dome (Figure S7). Temperature inversions are frequent in the High Arctic spring, with April mean depths of $660\,\mathrm{m}$ and $705\,\mathrm{m}$ at Alert and Eureka, Nunavut, respectively, and occurring on $75 - 85\%$ of days (Bradley et al., 1992). Owing to the static stability of the lower polar dome under these conditions, these air masses may be isolated from the air aloft and may be sensitive to different sources and

transport history (Stohl, 2006). Under these stable conditions, $CO$ and $CO_2$ were relatively constant (mean (quartile range), $144.5\,(144.2–146.5)\,\mathrm{ppb_v}$ and $405.8\,(405.4–406.2)\,\mathrm{ppm_v}$, respectively) in the lower polar dome and $O_3$ was depleted to $11.4$ $(3.1–23.4)\,\mathrm{ppb_v}$. Active halogen production and resulting $O_3$ depletion may occur largely at the surface (e.g., Spackman et al., 2010; Abbatt et al., 2012; Pratt et al., 2013). It follows that the observed $O_3$ profile could be interpreted as an indication of mixing of $O_3$ depleted air from the surface up to $\sim\!252\,\mathrm{K}$ ($\sim\!400\,\mathrm{m}$). Particle number concentrations between $100 - 500\,\mathrm{nm}$

($N_{100-500}$) were relatively constant in the lower polar dome ($\sim\!150\,\mathrm{cm^{-3}}$), while larger accumulation mode particles ($N_{>500}$) were most abundant in the lower polar dome compared to higher potential temperatures ($\sim\!4\,\mathrm{cm^{-3}}$ compared to $<\!1\,\mathrm{cm^{-3}}$). Second, in the middle polar dome ($252 - 265\,\mathrm{K}$), $O_3$ increased toward $\sim\!50\,\mathrm{ppb_v}$, $CO$ and $CO_2$ remained relatively constant while water vapour showed more variability. Finally, at the highest potential temperatures we observed more variability in $CO$, $CO_2$ and $H_2O$, while $O_3$ concentrations were relatively constant at $49.6\,(45.7–54.1)\,\mathrm{ppb_v}$. $N_{>500}$ was near zero in the upper

polar dome, while $N_{100-500}$ showed more variability compared to colder potential temperatures.

The importance of lower latitude source regions increases as potential temperature increases in the polar dome. FLEXPART-ECMWF potential emission sensitivities (Figure 3) indicate that air masses in the lower and middle polar dome had resided there for at least $10\,\mathrm{days}$, with significant sensitivity to the surface north of $80°\mathrm{N}$ and some sensitivity to high latitude Eurasia. The fraction of the previous 10 days spent in the polar dome is highest in the middle and lower polar dome, while above

$\sim\!265\,\mathrm{K}$ this quantity decreases significantly (Figure 4, S8). This observation indicates a clear separation in air mass history between the mid-to-lower polar dome and the upper polar dome. Sensitivity to lower latitude regions increases as potential temperature increases in the polar dome, particularly in high latitude Eurasia and North America (Figure 3). Locations of active fires during 28 March 2015 – 13 April 2015 and of oil and gas extraction emissions (Figure 3) indicate that biomass burning emissions likely had a stronger influence on the upper polar dome, while oil and gas extraction emissions may be

more important in the lower polar dome. Total March – May 2015 fire counts in the northern hemisphere were comparable to



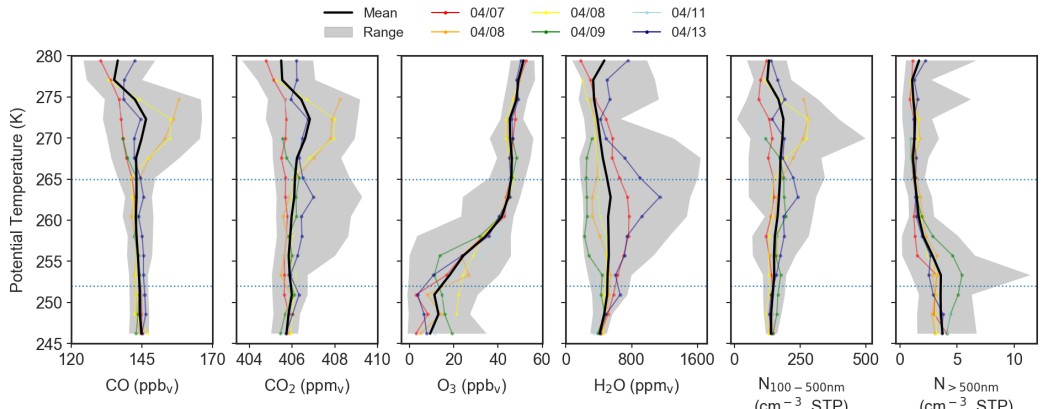

**Figure 2.** Mean potential temperature profiles of trace gases (CO, $CO_2$, $O_3$ and $H_2O$) and particle concentrations ($N_{100-500}$ and $N_{>500}$) in the polar dome observed during 7 – 13 April 2015. Coloured lines indicate the mean profile for each flight, the black line represents the mean profile over all flights, and gray shading shows the range of observations in each potential temperature bin. Horizontal dashed blue lines separate the lower, middle and upper polar dome defined as $245 - 252\,\mathrm{K}$, $252 - 265\,\mathrm{K}$ and $265 - 280\,\mathrm{K}$.

previous years (Figure S9), but were significantly lower than 2008. This suggests that biomass burning sources are often less important sources of Arctic aerosol than has been suggested by previous observations from the year 2008 (e.g., Warneke et al., 2009; Brock et al., 2011; Hecobian et al., 2011; McNaughton et al., 2011; Liu et al., 2015).

   A prevalent feature of air mass histories in the lower and middle polar dome is descent from aloft over at least 10 days
prior to our measurements (Figure 3g,h). The FLEXPART-ECMWF predicted plume centroid also shows some evidence for descent in the upper polar dome (Figure 3i), though we note that descent from aloft in the plume centroid does not preclude some sensitivity to the surface. Air mass descent in the polar dome is likely caused by a combination of both radiative cooling (on the order of $1\,\mathrm{K\,d^{-1}}$ (Klonecki et al., 2003)) and orographic effects over nearby elevated terrain on Ellesmere Island and Greenland. With long aerosol lifetimes under cold and relatively dry conditions in the polar dome, this suggests that aerosol in
the upper polar dome can influence the lower and middle polar dome on the time scale of 10 days and longer. Transport times to the Arctic lower troposphere are likely longer than 10 days (e.g., Brock et al., 2011; Qi et al., 2017; Leaitch et al., 2018), suggesting that a major springtime transport mechanism may be lofting near source regions followed by northward transport and descent into the polar dome (Stohl, 2006). In the next section, we discuss observed aerosol composition in the context of these transport patterns.

## 3.2 Aerosol composition in the polar dome

Vertical variability in aerosol composition was systematic across flights in the polar dome during April 2015. Sub-micron aerosol present in the coldest air masses of the lower polar dome contained the highest fraction of sulphate (74% by mass, Figure 5). This trend in the sulphate mass fraction ($mf_{SO4}$) was driven by both decreasing sulphate and increasing organic



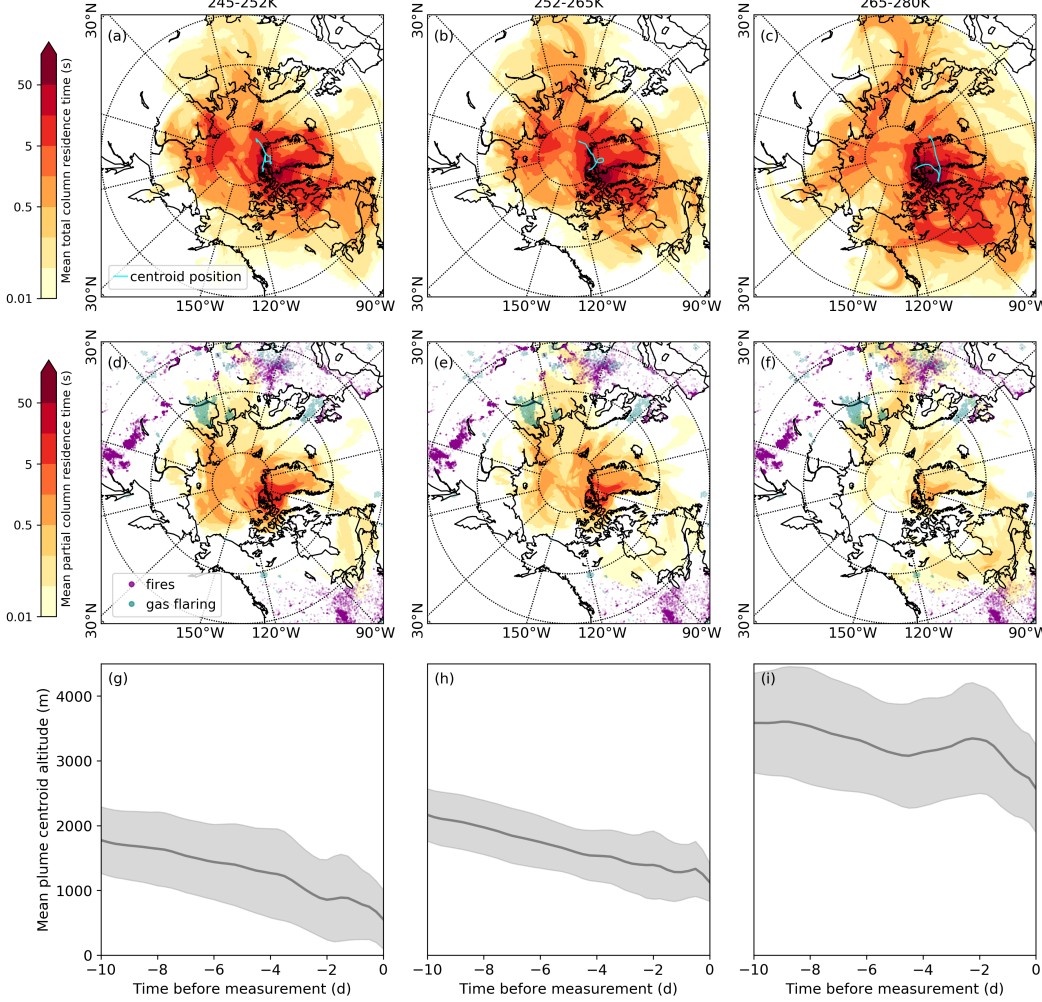

**Figure 3.** FLEXPART-ECMWF potential emission sensitivity (PES) and plume centroid altitude averaged over three potential temperature ranges in the polar dome. (a)–(c) Mean total column PES, (d)–(f), mean partial column (<200m) PES, (g)–(i) mean plume centroid altitudes for 245-252K (a,d,g), 252-265K (b,e,h) and 265-280K (c,f,i). Fire locations during 28 March 2015 to 13 April 2015 from MODIS are magenta points, gas flaring locations associated with oil and gas extraction from the ECLIPSE emission inventory (V5) for 2015 are light green points.

aerosol concentrations as potential temperature increased (Figure 6,S5). This observation is broadly consistent with previous vertically-resolved measurements of aerosol sulphate in both the Canadian Arctic and Alaskan Arctic during spring that have indicated increasing sulphate concentrations toward lower altitudes (Scheuer et al., 2003; Bourgeois and Bey, 2011). Large contributions of sulphate to near surface Arctic spring aerosol is also consistent with ground-based observations at long term

5   monitoring stations including Zeppelin, Svalbard; Alert, Nunavut; and Utqiaġvik (Barrow), Alaska (e.g., Barrie and Hoff,



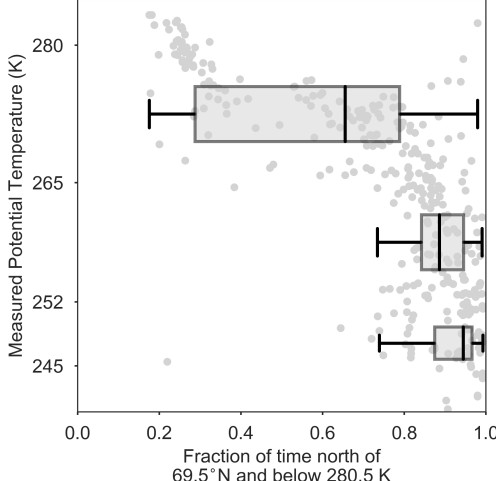

**Figure 4.** Observed potential temperature (K) as a function of FLEXPART-ECMWF predicted fraction of the past 10 days in the polar dome (i.e., below 280.5 K and north of 69.5°N). The FLEXPART-ECMWF relative residence time is binned in the lower (245 – 252 K), middle (252 – 265 K) and upper (265 – 280 K) polar dome.

1985; Quinn et al., 2007; Breider et al., 2017; Leaitch et al., 2018). The mass fraction of ammonium ($mf_{NH4}$) increases with increasing potential temperature. This trend is driven by both decreasing sulphate concentration and increasing ammonium concentration as potential temperature increases (Figure 6). This observation is broadly consistent with previous vertically-resolved measurements in the North American Arctic from April 2008 that demonstrated increased ammonium relative to

sulphate toward higher altitudes (Fisher et al., 2011). However, Fisher et al. (2011) observed significantly higher ammonium content compared to our measurements. These differences may arise from the larger altitude range in Fisher et al. (2011) (up to ~10 km) and differences in source regions or source strengths between 2008 and 2015.

Organic aerosol (OA) and refractory black carbon (rBC) were more abundant in the upper polar dome, while sulphate was less abundant. On average, OA and rBC contributed 42% and 2% to aerosol mass, respectively, in the upper polar dome. OA was

highly oxygenated throughout the polar dome, with oxygen-to-carbon (O/C) ratios above 0.5 in the majority of measurements (Figure S10). High O/C ratios are consistent with an abundance of highly functionalized organic acids observed in Arctic Haze aerosol at Alert, Nunavut during spring (Kawamura et al., 1996, 2005; Narukawa et al., 2008; Fu et al., 2009; Kawamura et al., 2010; Leaitch et al., 2018). Owing to the lack of unique mass spectral fragments from this highly oxygenated OA, our ToF-AMS spectra cannot distinguishing differences in OA composition in the polar dome. Overall, our observations suggest

that surface-based measurements may underestimate the contribution of OA, rBC and ammonium to aerosol transported to the Arctic troposphere in spring.

Aerosol composition varies systematically with model-predicted time spent in the middle to lower polar dome, where the time spent in the polar dome is the longest (Figure 4). The mass fractions of OA and rBC decrease with the FLEXPART-



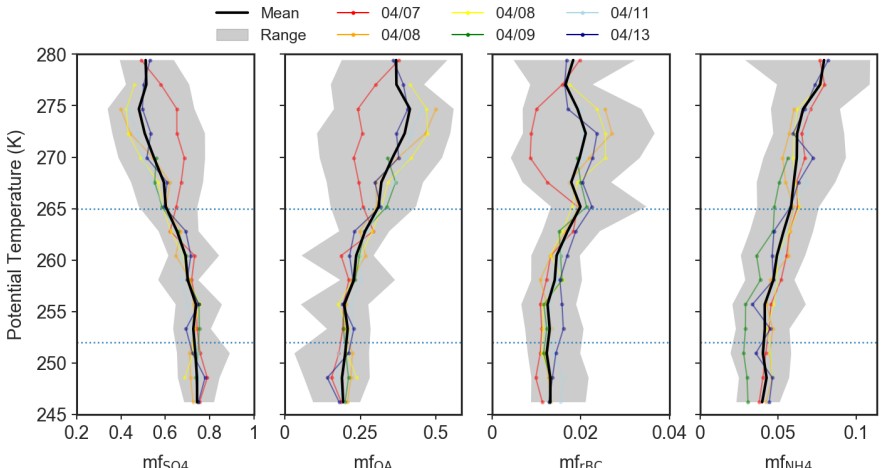

**Figure 5.** Mean potential temperature profiles of relative aerosol composition, including mass fractions of sulphate (mf$_{SO4}$), organic aerosol (mf$_{OA}$), refractory black carbon (mf$_{rBC}$), and ammonium (mf$_{NH4}$), in the polar dome observed during 7 – 13 April 2015. Coloured lines indicate the mean profile for each flight, the black line represents the mean profile over all six flights, and gray shading shows the range of observations in each potential temperature bin.

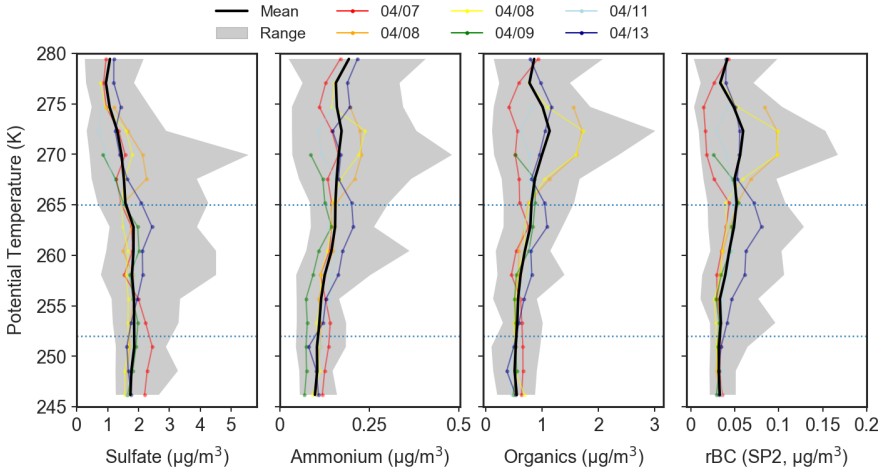

**Figure 6.** Mean potential temperature profiles of absolute (STP) sub-micron aerosol composition in the polar dome observed during 7 – 13 April 2015, including sulphate, organics and ammonium from the ToF-AMS and refractory black carbon (rBC) from the SP2. Nitrate concentrations were negligible, and largely below detection limits. Coloured lines indicate the mean profile for each flight, the black line represents the mean profile over all six flights, and gray shading shows the range of observations in each potential temperature bin.



ECMWF predicted fraction of the previous 10 days spent north of 69.5°N and below 265 K (Figure 7). OA and rBC were well-correlated in the middle and upper polar dome (Figure S11), suggesting that these species have a similar source region and/or have undergone similar processing. A dominance of anthropogenic (fossil fuel) sources of black carbon to the High Arctic during April 2015 may explain this relationship between rBC and OA. The importance of anthropogenic emissions of

black carbon from East and South Asia to measured Arctic black carbon in spring was recently demonstrated using a chemical transport model constrained by our measurements of black carbon in combination with surface sites and previous aircraft-based campaigns (Xu et al., 2017). European and North Asian anthropogenic emissions contributed significantly to Arctic black carbon in the lowest kilometre, with East and South Asian sources increasing in importance toward higher altitudes (Xu et al., 2017). South Asian regions are not well-represented in 10 day FLEXPART-ECMWF backward simulations, which likely do

not capture transport back to all source regions (Qi et al., 2017; Leaitch et al., 2018). OA and rBC are largely uncorrelated in the lower polar dome, suggesting shifting source regions and/or chemical processing of OA toward lower potential temperatures. This observation is consistent with multi-year observations from Alert, Nunavut showing that black carbon and organic matter are correlated during winter, but become uncorrelated during spring (Leaitch et al., 2018).

In contrast to OA and rBC, the mass fraction of sulphate increases with increasing time spent in the mid-to-lower polar dome

(Figure 7). In the upper polar dome the ammonium to sulphate molar ratio is at times consistent with ammonium bisulphate, while more sulphuric acid is likely present at lower potential temperatures. The enhanced fraction of sulphate in the lower polar dome compared to higher potential temperatures could arise from a combination of possible mechanisms. First, the stability of the polar dome may cause systematic vertical variability in source regions throughout the polar dome (e.g., Stohl, 2006). The observed middle-to-lower polar dome aerosol composition could arise from high latitude, sulphur-rich emissions in the

absence of significant ammonia and organic aerosol sources. A complex mixture of natural and anthropogenic sources has previously been shown to contribute to observed variations in sulphate and ammonium with altitude in Arctic spring (Fisher et al., 2011). In the lowest 2 km, emissions from non-Arctic Russia and Kazakhstan (included as part of East and Southern Asia in Xu et al. (2017)), along with North American, were the dominant sources of sulphate in April 2008 (Fisher et al., 2011). At higher altitudes, model results suggested East Asian sources of sulphate become more important and, along with European

sources, were the main contributor of sulphate aerosol in Arctic spring. Second, and possibly in addition to shifting source regions, aerosol composition could be changing as a result of chemical processing over the long aerosol lifetime. The fraction of sulphate could be increasing with decreasing potential temperature as a result of oxidation of transported sulphur dioxide and subsequent condensation of sulphuric acid onto existing particles as air masses slowly descend (Figure 3). In addition, oxidation of existing OA, resulting in fragmentation and loss of aerosol mass to the gas phase could contribute to a decrease in

OA concentrations toward colder potential temperatures. Descent from aloft appears to be an important transport mechanism influencing the lower polar dome in our flight area, lending some support to this second set of processes. Finally, wet removal or cloud processing of aerosol over long transport times likely impacts the aerosol composition we observe, though we cannot distinguish this influence with our measurements. In the next section, we examine the characteristics of lower polar dome aerosol in detail and compare it to aerosol present in the middle and upper polar dome.





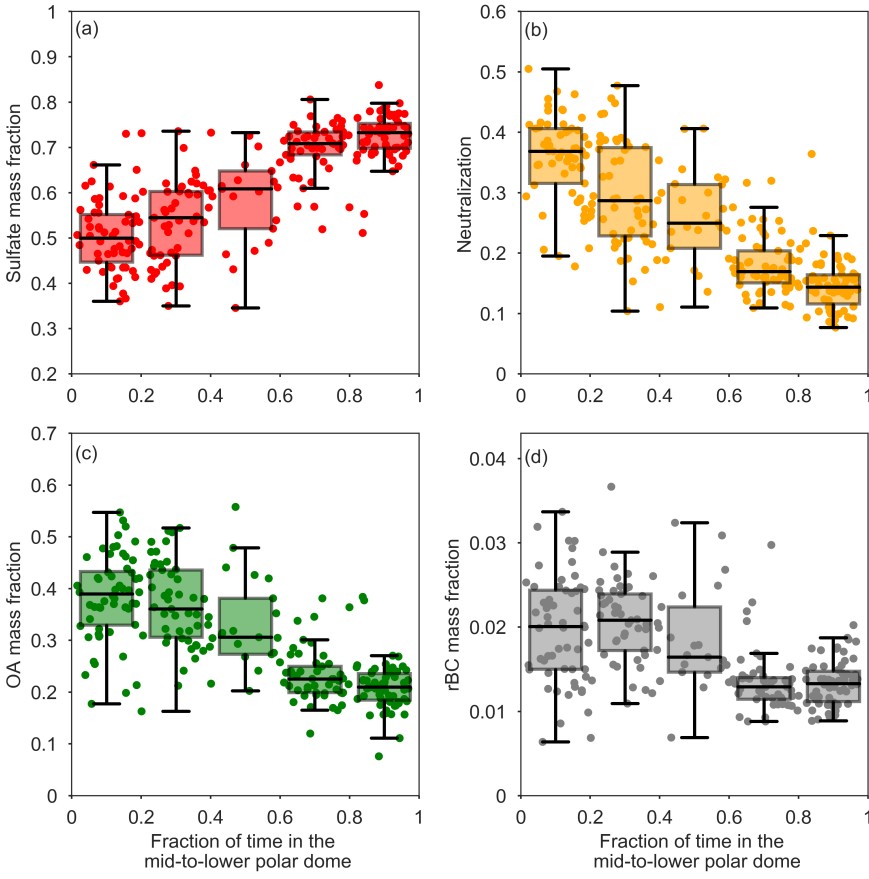

**Figure 7.** Submicron aerosol mass fractions as a function of FLEXPART-ECMWF-predicted fraction of the previous 10 days prior to measurement spent in the middle to lower polar dome (north of 69.5°N (Bozem et al., 2018) and, based on trace gas profiles, below 265 K (∼1600 m)).

## 3.3 Characteristics of lower polar dome aerosol

Lower polar dome air masses had resided for the longest times within the polar dome (Figure 3 and S8), suggesting that this aerosol likely had a lifetime of 10 days or longer. This aerosol was comprised largely of sulphate, with smaller amounts of OA, rBC and ammonium compared to aerosol present in the middle and upper polar dome (Section 3.2). In addition, ToF-AMS
5  spectra provide qualitative evidence for the presence of sea salt aerosol in the lower polar dome, which decreases to negligible concentrations through the middle polar dome (Figure 8). The ToF-AMS $NaCl^+$ signal and $N_{>500}$ have a similar profile, suggesting that sea salt may be associated with the increase in larger accumulation mode particles observed in the lower polar dome. This observation is consistent with previous airborne measurements in the Alaskan Arctic during spring that showed the largest fraction of sea salt particles were present in air masses identified as associated with the "Arctic Boundary Layer"



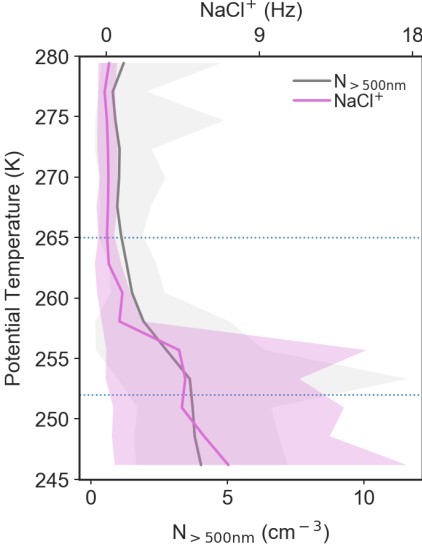

**Figure 8.** Potential temperature profiles of the ToF-AMS $NaCl^+$ signal (top axis), as a qualitative indication of the presence of sea salt aerosol, and $N_{>500}$ (bottom axis). The solid lines represents the mean profile for 7 – 13 April 2015, and shading represents the range of measurements in each potential temperature bin.

(i.e., identified by depleted $O_3$ concentrations) (Brock et al., 2011). Sea salt contributes significantly to aerosol observed at ground-based long term monitoring stations (e.g., Quinn et al., 2002; Leaitch et al., 2013; Huang and Jaegle, 2017; Leaitch et al., 2018), and peaks in concentration during winter to early spring. Sources of sea salt at high northern latitudes in spring include transport of sea salt from northern oceans, production of sea salt aerosol from open leads in sea ice (e.g., Leck et al.,

2002; Held et al., 2011; May et al., 2016), and production of saline aerosol through wind driven processes over ice and snow (Yang et al., 2008; Shaw et al., 2010; Xu et al., 2016; Huang and Jaegle, 2017). The strong decrease in $NaCl^+$ signal and $N_{>500}$ above the lower polar dome is suggestive of a near surface source of sea salt in the High Arctic; open leads or wind-driven ice and snow processes may contribute to lower polar dome aerosol. Recent observations at Utqiaġvik (Barrow), Alaska have demonstrated the prevalence of sea salt aerosol in Arctic winter and significant mixing with sulphate (Kirpes et al., 2018).

Sulphate may be internally mixed with sea salt in the lower polar dome; however, owing to low particle concentrations we are unable to obtain an $NaCl^+$ signal from size resolved mass spectra.

  Size-resolved observations of non-refractory aerosol composition provide evidence for different particle mixing states across the size distribution. On average, sulphate was present in larger particle sizes in the lower polar dome compared to the middle and upper polar dome (Figure 9). In contrast, OA size distributions were very similar in the lower and upper polar dome

(Figure S12). In the lower polar dome, the fraction of sulphate increases with particle size (Figure 9), implying the presence of different particle mixing states, and different particle sources or chemical processing in the polar dome. Single particle observations from two flights (Figure 10, Figure S13) are consistent with these bulk size resolved observations. Accumulation mode



particles were highly internally mixed, consistent with very aged particles, but the presence of sulphate-rich and organic-rich particles was discernible from cluster analysis of ToF-AMS spectra. Sulphate-rich particles were dominant in the coldest air masses and were larger in size compared to organic-rich particles (Figure 10 and S11). Increasing sulphate fraction toward larger particle sizes (Figure 9) suggests that sulphuric acid may have condensed on existing particles, growing them to larger

sizes. An increase in aerosol sulphate toward lower altitudes and a simultaneous decrease in gas phase $SO_2$ has been observed previously in Arctic spring; this could be consistent with oxidation of $SO_2$ and condensation on pre-existing particles in the lowest $1 - 2\,km$ (Bourgeois and Bey, 2011). Barrie and Hoff (1984) estimated a mean oxidation rate of $SO_2$ to sulphuric acid in April of $2.4 - 4.8\,\%\,d^{-1}$, which could explain the enhanced concentrations of sulphate toward lower potential temperatures in our observations. While the smaller OA size in the lower polar dome could be consistent with loss of OA mass through

fragmentation processes, similar OA size distributions in the lower and upper polar dome appear to negate this possibility (Figure S12). Our observations do not distinguish unambiguously between vertical variability in source composition, chemical processing during descent in the polar dome and wet removal or cloud processing during transport. All processes likely contribute to the systematic vertical variability in High Arctic aerosol composition that we observe.

## 4   Conclusions

In the Arctic spring polar dome, aerosol composition and trace gas concentrations varied systematically with potential temperature. We defined the lower $(245 - 252\,K)$, middle $(252 - 265\,K)$ and upper $(265 - 280\,K)$ polar dome based on vertical profiles of trace gases. The contribution of sulphate increased from the upper to lower polar dome (mean mass fractions 48% and 74%, respectively), while organic aerosol, refractory black carbon and ammonium were more abundant in the upper polar dome (mean mass fractions 42%, 2% and 8%, respectively). At the lowest potential temperatures, in the lower polar dome,

sulphate-rich particles were present at larger accumulation mode sizes compared to the upper polar dome. Our observations indicate that long-term, surface-based measurements may underestimate the contribution of organic aerosol, refractory black carbon and ammonium to aerosol transported to the High Arctic troposphere in spring. In addition, our observations of sea salt signals in the lower polar dome suggest that the significant sea salt concentrations observed at long-term monitoring stations in spring may not occur throughout the depth of the polar dome. These systematic differences in aerosol composition with

potential temperature likely arise through a combination of mechanisms. First, aerosol from different source regions, with differing composition, arrives at a range of potential temperatures in a stable atmosphere. Second, aerosol composition can be altered by chemical processing of transported aerosol and sulphur dioxide during descent into the polar dome over periods of 10 days or longer. Third, wet removal and cloud processing near emission and along the transport path may impact the composition of aerosol arriving in the polar dome, though this influence is difficult to distinguish with our observations. Modelled air

mass history from FLEXPART-ECMWF demonstrates that this systematic variation in aerosol composition is in part related to differing transport regimes as a function of potential temperature. In the lower polar dome, air masses had resided in the High Arctic region for at least 10 days prior to measurement and had largely descended from higher altitudes. Some sensitivity to the High Arctic surface could explain the observed sea salt in the lower polar dome. Lower latitude source regions in




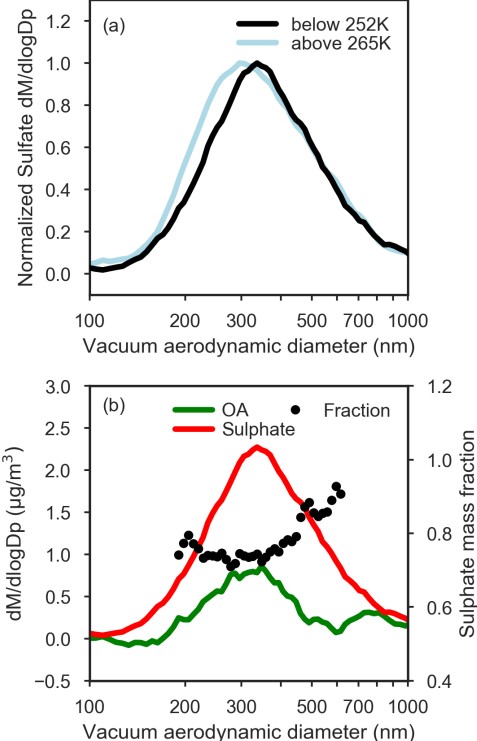

**Figure 9.** (a) Normalized mean ToF-AMS size distributions of sulphate subset by observed potential temperature: below 252 K (black), above 265 K (light blue). (b) ToF-AMS size distributions of sulphate (red) and total organic aerosol (green) below 252 K. The mass fraction of sulphate calculated from ToF-AMS size distributions is shown on the right axis in black circles, and is calculated only between 200 and 600 nm owing to low OA signals at smaller and larger sizes.

Europe, Asia and North America became more important toward higher potential temperatures in the upper polar dome. Long transport times make source diagnosis difficult using 10 day backward trajectories, and chemical processing during long Arctic residence times contributes to challenges in identifying source regions of lower polar dome aerosol. Using our observations, we cannot quantitatively distinguish the relative importance of vertical variability in source composition, chemical processing during descent in the polar dome and removal or cloud processing during transport. Our observations present a challenge to chemical transport models for their representation of the processes impacting High Arctic aerosol in spring.





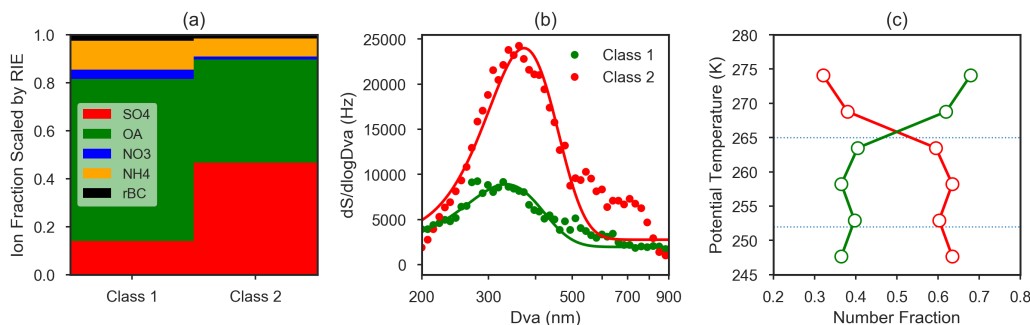

**Figure 10.** Summary of k-means cluster analysis of 1677 single particle (ETSP) spectra obtained on two flights (8 April and 13 April, 2015). (a) Bar chart of relative ion fraction scaled by the relative ionization efficiency (RIE) of each species, for two particle classes obtained by k-means cluster analysis. Particle class 1 is referred to as "organic-rich" and class 2 is referred to as "sulphate-rich." (b) Mean size distributions (expressed as dSignal/dlogDva, Hz) of the two particle classes (points) and gaussian fits to the observations (lines). (c) Mean relative abundance of class 1 (green, organic-rich) and class 2 (red, sulphate-rich) binned by potential temperature. Horizontal lines represent the divisions between the lower, middle and upper polar dome.





*Data availability.*   All data from NETCARE will be available on the Government of Canada Open Data Portal (https://open.canada.ca/data/en/dataset)
in collaboration with ECCC, until that time and for the model data please contact the NETCARE PI J. Abbatt (jabbatt@chem.utoronto.ca) for
data access. Sea ice data shown in Figure 1 is available at: https://seaice.uni-bremen.de/sea-ice-concentration/. Global MODIS active fire lo-
cations shown in Figure 3 are available at: https://earthdata.nasa.gov/earth-observation-data/near-real-time/firms/active-fire-data. Gas-flaring

locations from the ECLIPSE inventory V5 shown in Figure 3 are available at: http://www.iiasa.ac.at/web/home/research/researchPrograms/
air/ECLIPSEv5.html

*Author contributions.*   MDW wrote the manuscript, with significant conceptual input from DK, HB, WRL and JPDA, and critical feedback
from all co-authors. MDW, HB, JB, AKYL, HS and WRL operated instruments in the field and analysed resulting data. SH and AAA
analysed flight data. WRL, JPDA, ABH and AB designed the field experiment. DK ran FLEXPART simulations, and MDW analysed the

resulting data with input from DK and HB.

*Competing interests.*   The authors declare no competing interests.

*Acknowledgements.*   Funding for this work was provided by the Natural Sciences and Engineering Research Council of Canada (NSERC)
through the NETCARE project (www.netcare-project.ca) of the Climate Change and Atmospheric Research Program, the Alfred Wegener
Institute (AWI) and Environment and Climate Change Canada (ECCC). We gratefully acknowledge Kenn Borek Air Ltd, in particular our

pilots and crew Garry Murtsell, Neil Traverse and Doug Mackenzie, for their support of our measurements. Logistical and technical support
before and during the campaign was provided by a number of contributors; in particular by Desiree Toom-Sauntry (ECCC), Ralf Stae-
bler (ECCC), Katherine Hayden (ECCC), Andrew Elford (ECCC), Anne Marie MacDonald (ECCC), Maurice Watt (ECCC), Mohammed
Wasey (ECCC), Jason Iwachow (ECCC), Alina Chivulescu (ECCC), Ka Sung (ECCC), Dan Veber (ECCC), Julia Binder (AWI), Lukas
Kandora (AWI), Jens Herrmann (AWI) and Manuel Sellmann (AWI). Extensive logistical and technical support was provided by Andrew

Platt (ECCC), Mike Harwood (ECCC) and Martin Gerhmann (AWI). We are grateful to CFS Alert and Eureka Weather Station for sup-
porting the measurements presented in this work. Data was analysed in Igor Pro 6.37 (www.wavemetrics.com/index.html) and Python 3.5.2
(www.python.org/downloads/release/python-352/). We acknowledge Charles Brock, Jennifer Murphy and Dylan Jones for their comments
on an early version of this manuscript.



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
