# Peer review of "Aircraft-based measurements of High Arctic springtime aerosol show evidence for vertically varying sources, transport and composition"

_Atmospheric Chemistry and Physics, 2018_

## Referee Comment (RC1) · Anonymous Referee #1 · 3 Sep 2018

Willis et al describe aircraft measurements of aerosols during the NETCARE campaign in the high Arctic during springtime. The revisions made following the quick review have significantly improved the manuscript, and I only have minor suggested revisions here. These line numbers correspond to the track changes version of the manuscript. The title now reflects the unique aspects of this particular study, which is helpful, and inclusion of the new Figure 4 in the main text is helpful as well.

My main comments correspond to Page 9, Lines 6-7: What density is assumed to compare the ToF-AMS non-refractory aerosol with the UHSAS aerosol mass? Also, it is stated that the AMS mass and UHSAS mass were within a factor of two. Was the bias

consistently in one direction? Was there a meaningful temporal variation associated with the difference? A factor of two could mean that the AMS may have measured half or twice as much aerosol mass, which is a lot! This should be commented on. Was the discrepancy correlated with periods influenced by NaCl+, suggesting that the missing mass could be sea salt?

Minor Comments: Page 1, Lines 5-6: Remove reference.

Page 1, Line 8: What is the reason for listing ammonium as "NH4" instead of showing the 2+ charge and showing the 4 as subscript? This "NH4" formatting is used throughout.

Page 1, Lines 18-19: Also state in this sentence that these species are altitude-dependent. Surface-based measurements will reflect long-range transport at the surface, but not the total transport to the Arctic. Please clarify this sentence.

Page 2, Line 28: Add reference.

Page 11, Line 13: This is an old reference, and given the changing climate, it would seem that the temperature inversion frequency and depths may have changed over time. Is there a newer reference, or any evidence of recent change?

Page 11, Line 18: Add Oltmans et al 2012, JGR, which shows ozonesonde data for near-surface vertically-resolved ozone levels.

Page 14, Lines 6-7: For context, please provide the ammonium content in parentheses, or integrate in the sentence.

Figure 7: Please remove plot (b) "Neutralization", since this discussion has been removed per the quick review. The other three plots in this figure are very useful. Could the absolute concentrations be integrated in this figure, as well, perhaps as average markers?

---

## Referee Comment (RC2) · Anonymous Referee #3 · 24 Sep 2018

Summary: This very well-written paper presents vertical profiles of various atmospheric constituents (gas and aerosol phase) measured during an airborne field campaign in the Arctic. The authors utilize airmass history modelling data to suggest that long range transport brings in aerosols and gases from lower latitudes at high altitude and subsequently these transported species subside to lower levels of the atmosphere. Their conclusion is that relying solely on long-term, surface-based in-situ measurements in the Arctic may lead to underestimation of some species in the Arctic troposphere.

Science comments: Obviously, long-term surface measurements don't tell the whole story, particularly in the Arctic, but I do think it would be worthwhile for the authors to

reiterate towards the end of the manuscript that the surface measurements can provide value in terms of context/representativeness for at least lower altitude measurements. For example, as the authors note, 2008 was an anomalous year and not necessarily representative of Arctic climatology (P11, line34 –P12, line2). I realize the authors note this on P3, lines24-25, but could be noted again later.

Related to the comment above – many of the species measured on the aircraft are also measured at the surface at Alert. How do the surface measurements compare to the lowest potential temperature airborne measurements? Could those surface data (or some statistically appropriate summary of the surface data) be overlaid on the relevant plots (i.e., Figures 2, 4, and 6)? Something like this is done in Figure S4 for CO for the middle and lower polar dome, but it seems like the authors could place a symbol (or box-whisker) on the profile plots in the main manuscript showing the surface values of various parameters for April 2015 or April 7-13 2015 or the median over all Aprils or ...

I found the discussion of PES a bit confusing. PES is given in units of seconds (P10, line 3), but on page 11, line 26 the authors talk about PES indicating that airmasses had residence times of at least 10 days. I'm not sure how one goes from PES values on order of 50 s (scale in Fig 5) to 10 days. I didn't see a reference to explain this, so it'd be helpful if the authors could either provide a reference or explain the relationship a little more clearly. Perhaps the scale in Fig 5 is actual log(seconds)?

Technical/editing comments:

P9, line 34: providing a time-resolved information –> providing time-resolved information

P10, line 2: provide approximate altitude or pressure range the 10 vertical levels in FLEXPART-ECMWF correspond to. Are the vertical bins evenly spaced?

P11, line 28: observations indicates –> observation indicates

P13, Figure 3: give indication of latitude values, the outer circle is 30 N but what are

inner circles?

P14, line 13: cannot distinguishing–> cannot distinguish

P14, lines 16-17: first sentence of paragraph is unclear. I think it is missing something?

P15, Figure 5: say what boxes whiskers represent is it 5,25,50,75,95 percentiles or ?? and say represent dots in each 0.2 fraction of time interval.

P16, Figure 8: should 'gaussian' be capitalized?

———————————————————

---

## Referee Comment (RC3) · Anonymous Referee #4 · 1 Nov 2018

This paper describes a carefully analysed and well thought through interpretation of a complex aircraft data set that provides significant new insight into the transport pathways and composition of aerosol in the Arctic polar dome. The analysis sets the observations in the framework of potential temperature to best reflected the vertical distribution of layers entering and descending the lower polar troposphere in spring time. The authors use an analysis of gas phase species to discriminate different layers within the polar dome and then analyse the aerosol characteristics within those layers. This paper offers very valuable new information on aerosol in the polar springtime and contextualises the more extensive surface based measurements that extent. It is a very thorough study and one that adds important new information to aerosol characterisa-

tion of the Arctic atmosphere. The paper is clear and very well written and despite the necessary detail conveys the information succinctly and in a way that is accessible for the non expert in most places. The figures and tables are informative and there is the right balance between information in the main text and the supplementary material. I recommend acceptance in ACP. I do have a few detailed comments that the authors should take on board:

Page 1: Abstract: "These differences in transport history were closely related to aerosol composition" It would be preferable to say that "Variations in aerosol composition were closely related to these differences in transport history" since transport history drives aerosol composition not the other way around. Page 2: line 30: It is not obvious how the work in this paper has relevance to this problem since the flights were conducted at altitudes of 3.5 km or less. I would suggest removing this as a problem that this work can shed light on. Page 3 line 18: I would start a new paragraph with "Sea salt...." Page 3: line 21-29: The text states that input to the Arctic from both Europe and Asia has been decreasing in the past decade yet most of the papers cited are either model studies or inversions constrained by surface measurements it may be worth tempering the statement to reflect the comments that follow. Page 8 lines 11-15: The technical details of AMS sampling will be lost to many non AMS users. It would be good to include references here to guide a non expert who wishes to know and include a line that summarises what data is delivered from each mode. Page 8: lines 27-28: It would be good to understand how variable these calibrations were and whether there was any trend. Were the calibrations all used and was the average taken or an interpolation performed? Page 9 lines 5-6: Given the low concentrations and the reliance of the CDCE methodology being very dependent on the ability to retrieve accurate ammonium concentrations from the fragmentation table over a wide range of water concentrations it would be good to understand the size of any potential bias in the calculation and well as the variation over the flight window. Page 9 line 21 and line 28: Figure S2 should read Figure S3 Page 9 line 32: strictly Centre not Center Page 10 line 25: demonstrate(s) Page 11 line 30: The caption and axes labels

in Figures 4 and S8 appear to be the same but the data are different. This needs clarification. Page 12 line 3: This statement agrees with Liu et al 2015 but is in contrast with the other papers. This might be made more clear by rewording. Page 14 line 5: Could these differences also be a result of the difficulties in retrieving ammonium from the MS at such low concentrations? Page 16 lines 28-30: This process would need to drive against an opposing temperature gradient driving condensation with reducing altitude though. Previous work suggests this highly aged OA is of low volatility. This might be worth commenting on. Page 18 figure 8 caption "...solid lines represent..." Page 20 figure 9: The ToF signals of organic aerosol at sizes above 500 nm are likely to have very poor counting statistics. The lack of oscillation of the ToF signal at high sizes also suggests the baseline is incorrectly accounted for. I am somewhat sceptical that the increase in OA/SO4 above 400 nm to 600 nm can be observed above the signal to noise. I would like to see the Poisson statistics and an assessment of the baseline correction before much claim is made for this enhancement. Page 19 lines 19-22: Conclusions: is the converse true? That is do surface based observations overplay the contribution of export of sulfate to Arctic in spring compared to the aircraft measurements presented here? If so it is worth stating. Pages 20 and 21: figures 9 and 10: can you explain the differences between the sulfate size distributions in figures 9 and 10?
* * *

---

## Author Comment (AC1) · 10 Dec 2018

We thank the reviewers for their comments that have improved the completeness and clarity of this manuscript. Our responses to comments and the corresponding changes to the manuscript are detailed below in blue text.

**Responses to Reviewer #1**

Willis et al describe aircraft measurements of aerosols during the NETCARE campaign in the high Arctic during springtime. The revisions made following the quick review have significantly improved the manuscript, and I only have minor suggested revisions here. These line numbers correspond to the track changes version of the manuscript. The title now reflects the unique aspects of this particular study, which is helpful, and inclusion of the new Figure 4 in the main text is helpful as well.

My main comments correspond to Page 9, Lines 6-7: What density is assumed to compare the ToF-AMS non-refractory aerosol with the UHSAS aerosol mass? Also, it is stated that the AMS mass and UHSAS mass were within a factor of two. Was the bias consistently in one direction? Was there a meaningful temporal variation associated with the difference? A factor of two could mean that the AMS may have measured half or twice as much aerosol mass, which is a lot! This should be commented on. Was the discrepancy correlated with periods influenced by NaCl+, suggesting that the missing mass could be sea salt?

We agree entirely that this issue merits further comment. Given the low aerosol mass concentrations and the prevalence of sea salt at low altitudes, it is challenging to arrive at perfect agreement between the ToF-AMS and other particle instruments. Two main issues arose when comparing ToF-AMS particle mass with that estimated from other, size-resolved measurements of particle concentration. First, the AMS underestimates aerosol mass in the presence of sea salt. Second, when sea salt was not present, the AMS gives a higher mass concentration than that predicted by the UHSAS by approximately a factor of two. A similar discrepancy between AMS aerosol mass and UHSAS estimated mass concentrations was discussed in Willis et al. (2017) for measurements from NETCARE 2014. The source of this discrepancy is discussed further here, in the revised Section 2.1.4 and in the revised supplementary material.

In addition to the two issues stated above, it appears that the UHSAS may not be counting the very low concentrations of larger particles accurately, and giving approximately a factor of two lower particle numbers above 500 nm compared to an optical particle counter (OPC) that was also sampling during the NETCARE flights (Figure AC1). Unfortunately the OPC was not integrated onto the Polar6 for NETCARE 2014, and so is not included in the instrumental comparisons described by Leaitch et al. (2016) and a quantitative laboratory comparison between the UHSAS and OPC is not available. However, the OPC generally agreed well with large particle concentrations obtained from a Forward Scattering Spectrometer Probe (FSSP) Model 300 that was under-wing mounted. The OPC periodically provides spuriously high concentrations of particles between 250 - 400 nm and so we restrict the comparison to sizes above 500 nm.

Total particle concentrations above 4 and 10 nm were measured with two TSI condensation particle counters (CPCs). Total number concentrations from the CPCs and the number of particles larger than 85 nm from the UHSAS agreed reasonably because the number concentrations of Arctic haze particles are dominated by particles larger than 85 nm. Thus, there is no indication of a significant bias in the particle number concentrations measured with the UHSAS. The discrepancy between the OPC and UHSAS for particles larger than 500 nm can still happen as the number of

[Figure]

Figure AC1: Comparison between the number of particles greater than 500 nm measured by the UHSAS and OPC. (a) For all observations, average to 10 second time intervals. (b) An example time series comparison for Flight 4 on 8 April 2015.

particles larger than 500 nm make very small relative contributions to the total particle number concentration. While the number concentrations of particles above 500 nm are small, they contribute significant amounts of mass (e.g., up to $\sim 1 \text{ug/m}^3$ depending on the assumed density). For the above reasons, we have used UHSAS observations between $100 - 500$ nm and OPC observations above 500 nm. Please note that this is now stated clearly in Section 2.1.4, the previous version of this manuscript was in error by not making this adequately clear.

Assuming a particle density of 1.5 g/cm$^3$, we compared total non-refractory aerosol mass from the ToF-AMS with the mass estimated from combined data from the UHSAS and OPC (i.e., the number of particles between $100 - 500 \text{ nm}$ ($N_{100-500}$) derived from UHSAS observations and the number greater than $500 \text{ nm}$ ($N_{>500}$) from the OPC (Figure AC3)). This comparison illustrates that the ToF-AMS measurements do not reflect the mass concentrations measured by the UHSAS and OPC in the presence of sea salt. The ToF-AMS also overestimates aerosol mass compared to the OPC and UHSAS by approximately a factor of two when sea salt is not present. This discrepancy likely arises from biases in both the UHSAS and ToF-AMS measurements.

The UHSAS may underestimate particle number concentrations at sizes smaller than 500 nm, and while this underestimate is not significant enough to impact comparison with total number concentrations from the CPCs, such an underestimate could still impact particle volume. Comparison of the UHSAS volume size distributions with that measured at the Alert Observatory was possible for short periods when the aircraft flew low in the area (i.e., between 60-200 m above ground level). We include once such comparison in Figure AC4, which suggests the UHSAS underestimated particle volume between 250 nm and 500 nm, by about 20% in this case.

Owing to the discrepancies discussed above we have stated in Section 2.1.4 and 2.1.5 that the particle mass concentrations presented in this work should be treated with caution. However, we believe that these discrepancies do not prevent us from using these data to make the interpretations described in this manuscript, which rely particularly on relative changes in aerosol composition. In section 2.1.4 we now state: "ToF-AMS total non-refractory aerosol mass correlated well with estimated aerosol mass from the UHSAS and OPC, but was generally higher by approximately a factor of two (Figure S4, assuming a mean density of $1.5 \text{ g cm}^{-3}$). An important exception to this

[Figure]

Figure AC2: Comparison between the number of particles greater than 10 nm measured by the CPC and the number of particles greater than 85 nm measured by the UHSAS, averaged to 10 second time intervals. (a) For all observations when the difference in concentration measured by the two CPCs was negligible. (b) An example time series comparison for Flight 3 on 8 April 2015.

observation occurred when the ToF-AMS measured significant NaCl$^+$; at these times, the ToF-AMS total aerosol mass was relatively constant, indicating that sea salt was an important contributor to aerosol mass. These discrepancies are discussed further in Section 1 of the Supplement. Owing to the discrepancies between measured and estimated particle mass, we emphasize that absolute mass concentrations presented in this work should be treated with caution; however, these discrepancies do not prevent a useful interpretation of the ToF-AMS data based upon relative changes in particle composition."

[Figure]

Figure AC3: Comparison between ToF-AMS total non-refractory aerosol mass and aerosol mass estimated from the UHSAS and OPC particle size distributions, assuming a particle density of 1.5 $g/cm^3$ and coloured by (a) sampling altitude and (b) ToF-AMS uncalibrated signal for sea salt.

[Figure]

Figure AC4: Comparison between particle volume distributions measured by the UHSAS aboard Polar6 and measured with an SMPS at the Alert Observatory for periods during 7 April 2015 when the aircraft flew between $60 - 200$ m.

Minor Comments:

Page 1, Lines 5-6: Remove reference.

This reference has been removed from the abstract.

Page 1, Line 8: What is the reason for listing ammonium as "NH4" instead of showing the 2+ charge and showing the 4 as subscript? This "NH4" formatting is used throughout.

The aerosol mass spectrometry community has developed the convention of using the notation "NH4" for ToF-AMS peaks that correspond to ammonium ($NH_4^+$), and similarly "SO4" for peaks corresponding to sulphate species (i.e., $(NH_4)_2SO_4$, $NH_4HSO_4$, $H_2SO_4$) and "NO3" for peaks corresponding to nitrate species ($NH_4NO_3$, $HNO_3$). For clarity, we have removed "NH4" from the abstract. This notation is now introduced in the methods section, and used when referring to the mass fraction of measured species (e.g., $mf_{NH4}$, $mf_{SO4}$).

Page 1, Lines 18-19: Also state in this sentence that these species are altitude dependent. Surface-based measurements will reflect long-range transport at the surface, but not the total transport to the Arctic. Please clarify this sentence.

Our intention here was to highlight that surface-based measurements do not necessarily represent aerosol transported through the depth of the Arctic troposphere. We have rephrased this sentence as follows: "Our observations highlight the differences in Arctic aerosol chemistry observed at surface-based sites and the aerosol transported throughout the depth of the Arctic troposphere in spring."

Page 2, Line 28: Add reference.

References to Blanchet et al., 1994 and 1995 have been added to this sentence.

Page 11, Line 13: This is an old reference, and given the changing climate, it would seem that the temperature inversion frequency and depths may have changed over time. Is there a newer reference, or any evidence of recent change?

We had originally chosen this reference because it focused specifically on Alert and Eureka, NU, and was a nice illustration of temperature inversion frequencies at those location. However, we do agree that this is an old reference. We have added reference to newer work, including satellite-based climatologies, and edited the corresponding sentence to be more general, as follows: "Temperature inversions are frequent in the High Arctic spring, with median inversion strengths of $\sim 5 - 10\,K$ occurring frequently in March, April and May [Bradley et al., 1992; Tjernström et al., 2009; Zhang et al., 2011; Devasthale et al., 2016]." There may be evidence for long-term trends in temperature inversion strength and depth, but we believe that discussion of trends is beyond the scope of this paper.

Page 11, Line 18: Add Oltmans et al 2012, JGR, which shows ozonesonde data for near-surface vertically-resolved ozone levels.

This reference has been added.

Page 14, Lines 6-7: For context, please provide the ammonium content in parentheses, or integrate in the sentence.

Fisher et al., 2011 reports aerosol neutralization as $[NH_4^+]/(2\,[SO_4^{2-}]^+[NO_3^-])$ along with absolute concentrations of ammonium and sulphate, but does not report other aerosol species such as organic aerosol and black carbon. This statement is based on the fact that Fisher et al observe

more ammonium relative to sulphate compared to our measurements, but direct comparison of ammonium content (e.g., $mf_{NH4}$) is difficult so we do not include specific values. We have re-phrased this sentence to clarify that Fisher et al., 2011 observed higher ammonium relative to sulphate compared to our measurements.

Figure 7: Please remove plot (b) "Neutralization", since this discussion has been removed per the quick review. The other three plots in this figure are very useful. Could the absolute concentrations be integrated in this figure, as well, perhaps as average markers?

This plot should have been replaced with the corresponding $mf_{NH4}$ plot following the quick review, and the neutralization plot appeared in error. We have corrected this and part (b) of Figure 7 now shows $mf_{NH4}$.

**Responses to Reviewer #2**

Summary: This very well-written paper presents vertical profiles of various atmospheric constituents (gas and aerosol phase) measured during an airborne field campaign in the Arctic. The authors utilize airmass history modelling data to suggest that long range transport brings in aerosols and gases from lower latitudes at high altitude and subsequently these transported species subside to lower levels of the atmosphere. Their conclusion is that relying solely on long-term, surface-based in-situ measurements in the Arctic may lead to underestimation of some species in the Arctic troposphere.

Science comments: Obviously, long-term surface measurements don't tell the whole story, particularly in the Arctic, but I do think it would be worthwhile for the authors to reiterate towards the end of the manuscript that the surface measurements can provide value in terms of context/representativeness for at least lower altitude measurements. For example, as the authors note, 2008 was an anomalous year and not necessarily representative of Arctic climatology (P11, line34 – P12, line2). I realize the authors note this on P3, lines 24–25, but could be noted again later.

We agree entirely that surface-base measurements are incredibly important, and in no way meant to devalue these observations. To make this clear we have made the following two changes at the beginning and end of the manuscript. First, we have re-worded the final sentence of the abstract as follows: " Our observations highlight the differences in Arctic aerosol chemistry observed at surface-based sites and the aerosol transported throughout the depth of the Arctic troposphere in spring." Second, we have amended the conclusions section as follows: "While observations at long-term monitoring stations provide the majority of our knowledge about Arctic aerosol, decoupling of air masses near the surface from the rest of the polar dome means that surface-based observations may not represent the altitude dependent composition of aerosol transported to the Arctic troposphere. Our observations indicate that long-term, surface-based measurements may underestimate the contribution of organic aerosol, refractory black carbon and ammonium to aerosol transported to the High Arctic troposphere in spring. In addition, our observations of sea salt signals in the lower polar dome suggest that the significant sea salt concentrations observed at long-term monitoring stations in spring may not occur throughout the depth of the polar dome."

Related to the comment above – many of the species measured on the aircraft are also measured at the surface at Alert. How do the surface measurements compare to the lowest potential temperature airborne measurements? Could those surface data (or some statistically appropriate summary of the surface data) be overlaid on the relevant plots (i.e., Figures 2, 4, and 6)? Something like this is done in Figure S4 for CO for the middle and lower polar dome, but it seems like the authors could place a symbol (or box-whisker) on the profile plots in the main manuscript showing the surface values of various parameters for April 2015 or April 7-13 2015 or the median over all Aprils or ...

Offline measurements of ambient particle composition were made at Alert during the time of the NETCARE 2015 campaign. In particular, one sample from MacDonald et al., ACP, 2017 spans April 6 – 13, 2015. We have added the corresponding measurements of sulphate, ammonium and refractory black carbon to Figure 6. No measurement of organic aerosol mass is available from Alert at this time so we cannot compare mass fractions as shown in Figure 5.

I found the discussion of PES a bit confusing. PES is given in units of seconds (P10, line 3), but on page 11, line 26 the authors talk about PES indicating that airmasses had residence times of at

least 10 days. I'm not sure how one goes from PES values on order of 50 s (scale in Fig 5) to 10 days. I didn't see a reference to explain this, so it'd be helpful if the authors could either provide a reference or explain the relationship a little more clearly. Perhaps the scale in Fig 5 is actual log(seconds)?

Two concepts need to be clarified to make this issue clearer, and we describe this briefly here followed by changes to the manuscript. One is the PES value and the other is the PES distribution. The PES value in a particular grid cell (air volume) is the response function of a source-receptor relationship. The PES value is proportional to the particle residence time in that air volume and measures the simulated concentration at the receptor (i.e., the release point) that a source of unit strength in that air volume has for an inert tracer. This gives a value in seconds.

The conclusion that air masses resided for 10 days in the lower and middle polar dome comes from the PES distribution showing that most of the PES values are evident at high latitudes and not at mid-latitudes. This tells us that most air masses spend their time in the Arctic, instead of being recently transported to the Arctic.

We have expanded the description of PES in Section 2.2 as follows: "...In backward mode, the model provides an emission sensitivity function called the potential emission sensitivity (PES). The PES in a particular grid cell, or air volume, is the response function of a source-receptor relationship, and is proportional to the particle residence time in that grid cell [Hirdman et al., 2010]. PES values can be combined with emission distributions to calculate receptor concentrations, assuming the species is inert; however, we use the PES directly and show maps of PES with units of seconds (i.e., proportional to air mass residence time). Absolute residence times depend on the model output time step and the extent of spatial averaging. Maps of PES represent integration of model output over a period of time prior to sampling (i.e., 10 days), also referred to as the "time before measurement," and over a vertical range. We show maps of both the total column PES (i.e., 0 – 20 km) and partial column PES (i.e., 0 – 200 m), as emissions near the surface are of particular interest." We have also clarified wording in Section 3.1 to make it clear that conclusions about air masses having spent ∼10 days in the Arctic are derived from PES distributions.

Technical/editing comments:

P9, line 34: providing a time-resolved information – providing time-resolved information
The typo has been corrected.

P10, line 2: provide approximate altitude or pressure range the 10 vertical levels in FLEXPART-ECMWF correspond to. Are the vertical bins evenly spaced?
This sentenced has been amended as follows: "FLEXPART-ECMWF output was provided every three hours over the 10 day period, with horizontal grid spacing of 0.25° and 10 vertical levels (50, 100, 200, 500, 1000, 2000, 4000, 6000, 8000 and 10000 m)."

P11, line 28: observations indicates – observation indicates
The typo has been corrected.

P13, Figure 3: give indication of latitude values, the outer circle is 30 N but what are inner circles?
The locations of parallels illustrated on Figure 3 (and Figure 1) are now indicated in the figure captions.

P14, line 13: cannot distinguishing – cannot distinguish

This typo has been corrected.

P14, lines 16-17: first sentence of paragraph is unclear. I think it is missing something?

This sentence has been re-phrased for clarity as follows: "Air masses spent the longest times in the middle to lower polar dome (Figure 4), and aerosol composition varied systematically with time spent in this portion of the polar dome."

P15, Figure 5: say what boxes whiskers represent is it 5,25,50,75,95 percentiles or ?? and say represent dots in each 0.2 fraction of time interval.

Clarification has been added to the figure caption as follows: "Data points corresponding to individual FLEXPART releases are shown as circles, and summary statistics are shown as boxes ($25^{th}$, $50^{th}$, $75^{th}$ percentiles) and whiskers ($5^{th}$, $95^{th}$ percentiles) for data binned by time spent in the middle and lower polar dome."

P16, Figure 8: should 'gaussian' be capitalized?

This typo has been corrected.

**Responses to Reviewer #3**

This paper describes a carefully analysed and well thought through interpretation of a complex aircraft data set that provides significant new insight into the transport pathways and composition of aerosol in the Arctic polar dome. The analysis sets the observations in the framework of potential temperature to best reflected the vertical distribution of layers entering and descending the lower polar troposphere in spring time. The authors use an analysis of gas phase species to discriminate different layers within the polar dome and then analyse the aerosol characteristics within those layers. This paper offers very valuable new information on aerosol in the polar springtime and contextualises the more extensive surface based measurements that extent. It is a very thorough study and one that adds important new information to aerosol characterisation of the Arctic atmosphere. The paper is clear and very well written and despite the necessary detail conveys the information succinctly and in a way that is accessible for the non expert in most places. The figures and tables are informative and there is the right balance between information in the main text and the supplementary material. I recommend acceptance in ACP. I do have a few detailed comments that the authors should take on board:

Page 1: Abstract: "These differences in transport history were closely related to aerosol composition" It would be preferable to say that "Variations in aerosol composition were closely related to these differences in transport history" since transport history drives aerosol composition not the other way around.

    This sentence has been revised as suggested.

Page 2: line 30: It is not obvious how the work in this paper has relevance to this problem since the flights were conducted at altitudes of 3.5 km or less. I would suggest removing this as a problem that this work can shed light on.

    During the campaign, temperatures below -25°C were frequently observed below 3.5 km. As well, the sampled air masses may experience colder conditions at a later time. And so, we do believe the measurements are relevant to discussion of both homogeneous and heterogeneous ice nucleation. In particular, particles containing mineral dust, organic species, sea salt and neutralized sulphate can act as ice nuclei at temperatures both below and above the water homogeneous freezing temperature. To clarify this issue we have revised this sentence as follows: "Further, particles containing mineral dust, organic species, sea salt or neutralized sulphate can act as ice nuclei and increase ice crystal number, also leading to impacts..."

Page 3 line 18: I would start a new paragraph with "Sea salt..."

    We have separated this long paragraph starting with the following sentence: "Arctic Haze observed near the surface is largely acidic sulphate, with lesser contributions from organic aerosol, dust, nitrate, ammonium and sea salt."

Page 3: line 21-29: The text states that input to the Arctic from both Europe and Asia has been decreasing in the past decade yet most of the papers cited are either model studies or inversions constrained by surface measurements it may be worth tempering the statement to reflect the comments that follow.

    We agree entirely that these conclusions are based largely on modeling efforts. These efforts, constrained by observations and using a variety of modeling approaches, provide the best information we can access about the origins of Arctic aerosol.

Page 8 lines 11-15: The technical details of AMS sampling will be lost to many non AMS users. It would be good to include references here to guide a non expert who wishes to know and include a line that summarises what data is delivered from each mode.

While we agree that these details may not be useful to all readers, it is nonetheless important to include them for completeness. We have added reference to DeCarlo et al., 2006 and Onasch et al., 2012 here.

Page 8: lines 27-28: It would be good to understand how variable these calibrations were and whether there was any trend. Were the calibrations all used and was the average taken or an interpolation performed?

Calibrations for ammonium nitrate ionization efficiency (IE) produced similar results during the campaign as well as before and after, within $\sim$20%. Most importantly, the ratio of the ionization efficiency to the air beam (nitrogen, m/z 28) signal remained constant (to within $\sim$10%) during the campaign. Data were calibrated based on the most recent calibration and the measured air beam signal was used to account for any differences in sensitivity that might have arisen between flights.

Page 9 lines 5-6: Given the low concentrations and the reliance of the CDCE methodology being very dependent on the ability to retrieve accurate ammonium concentrations from the fragmentation table over a wide range of water concentrations it would be good to understand the size of any potential bias in the calculation and well as the variation over the flight window.

We agree that the CDCE relies heavily on our ability to accurately quantify ammonium concentrations relative to sulphate and other species. However, given that we are using an H-ToF AMS instrument we do not agree that these values rely heavily on the fragmentation table. We are able to near baseline resolve peaks for ammonium and water at m/z 16, 17 and 18, and our ability to quantify ammonium relies therefore on fitting these peaks and not estimating contributions from water. While it is true that with a unit mass resolution instrument these effects would produce a highly variably CDCE, this was not observed in our analysis. As stated on Page 9, line 5, the median (quartile range) collection efficiency correction was 18% (12 – 28%), with this variation being driven almost entirely by differences in composition with altitude.

Page 9 line 21 and line 28: Figure S2 should read Figure S3

The references to Figure S3 on line 28, Page 9 has been corrected. The figure reference to Figure S2 at line 21 is correct. Note that the supplemental figure numbers have changed in the revise version of the manuscipt.

Page 9 line 32: strictly Centre not Center

This typo has been corrected.

Page 10 line 25: demonstrate(s)

We believe "illustrates" is a reasonable usage here.

Page 11 line 30: The caption and axes labels in Figures 4 and S8 appear to be the same but the data are different. This needs clarification.

While we agree that there a close similarities between these two figures that could be confusing, the two figures do not have the same axes labels. The x-axis in Figure 4 is "Fraction of time north of 69.5° and below **280.5K**." In other words, and as stated in the figure caption this corresponds

to the fraction of time spent in the polar dome (i.e., all parts). The x-axis in Figure S8 is "Fraction of time north of 69.5° and below **265K**." In other words, and as stated in the figure caption this corresponds to the fraction of time spent in the middle and lower polar dome.

Page 12 line 3: This statement agrees with Liu et al 2015 but is in contrast with the other papers. This might be made more clear by rewording.

It is true that this observation agrees broadly with Liu et al 2015, and is in some contrast with other papers such as those cited on Page 12 line 7. These papers are largely from the IPY missions in 2008, when biomass burning was an important source seemingly at all altitudes sampled. We believe that our meaning is made clear in lines 5-6, which state that our observations suggest that biomass burning may not be the most important contributor to Arctic aerosol at all altitudes in all years.

Page 14 line 5: Could these differences also be a result of the difficulties in retrieving ammonium from the MS at such low concentrations?

Please see our response to the above comment focusing on the quantification of ammonium with the high resolution ToF-AMS used in these flights.

Page 16 lines 15-16: This process would need to drive against an opposing temperature gradient driving condensation with reducing altitude though. Previous work suggests this highly aged OA is of low volatility. This might be worth commenting on.

We certainly agree that volatilization from organic aerosol fragmentation processes would be countered somewhat at lower temperatures. We have revised this sentence as follows: "In addition, oxidation of existing OA, resulting in fragmentation and loss of aerosol mass to the gas phase could contribute to a decrease in OA concentrations toward lower altitudes [e.g., Kroll et al., 2009]; however, this process may be less important at low temperatures."

Page 18 figure 8 caption "...solid lines represent..."

This typo has been corrected.

Page 20 figure 9: The ToF signals of organic aerosol at sizes above 500 nm are likely to have very poor counting statistics. The lack of oscillation of the ToF signal at high sizes also suggests the baseline is incorrectly accounted for. I am somewhat skeptical that the increase in OA/SO4 above 400 nm to 600 nm can be observed above the signal to noise. I would like to see the Poisson statistics and an assessment of the baseline correction before much claim is made for this enhancement.

We agree entirely that the particle time of flight data suffers from low signal to noise ratios owing to low aerosol concentrations. This is generally the case for pToF data collected aboard aircraft and is the reason why we focus on the mean values over several flights. The baseline correction was done using a baseline region at particle flights times longer than $5000\mu s$, which corresponds to particle sizes above $2\mu m$ where no particle or air signals are present in the spectrum. The baseline correction cannot account for noise in the organic aerosol pToF spectrum resulting from low particle concentrations. We have added shaded errors to Figure 9, which correspond to the mean size distribution for sulphate and organics plus or minus one standard deviation. This highlights the uncertainty in pToF distributions and the corresponding uncertainty in the derived fraction of

sulphate as a function of size. In the caption of Figure 9 we state: "Shading corresponds to plus or minus one standard deviation for sulphate and organic aerosol size distributions, and the relatively large variation in size-resolved composition indicates that the derived mass fraction of sulphate as a function of size is uncertain." We note that similar conclusions were drawn from both single particle and bulk size resolved data, supporting these results despite the low signal to noise ratios in measurements of size-resolved composition.

Page 19 lines 19-22: Conclusions: is the converse true? That is do surface based observations over-play the contribution of export of sulfate to Arctic in spring compared to the aircraft measurements presented here? If so it is worth stating.

The converse is likely true, and we have revised this sentence to reflect this: "Our observations indicate that long-term, surface-based measurements may underestimate the contribution of organic aerosol, refractory black carbon and ammonium to aerosol transported to the High Arctic troposphere in spring, while overestimating the contribution of sulphate."

Pages 20 and 21: figures 9 and 10: can you explain the differences between the sulfate size distributions in figures 9 and 10?

Differences in the suphate size distribution between Figures 9 and 10 arise from different sampling methods (i.e., bulk particle time-of-flight, pToF, versus single particle), as well as different sampling periods. Due to instrument software restrictions during this campaign, the bulk pToF and single particle modes could be be run at the same time. The single particle data was collected on two flights and therefore represents less averaging compared to the bulk pToF data. The single particle data is also shown for all altitudes, whereas the bulk pToF data is shown separated by altitude, illustrating differences in the sulphate particle size.